# Noise-induced plasticity of KCNQ2/3 and HCN channels underlies vulnerability and resilience to tinnitus

**Shuang Li, Bopanna I Kalappa, Thanos Tzounopoulos\***

Departments of Otolaryngology and Neurobiology, University of Pittsburgh School of Medicine, Pittsburgh, United States

**Abstract** Vulnerability to noise-induced tinnitus is associated with increased spontaneous firing rate in dorsal cochlear nucleus principal neurons, fusiform cells. This hyperactivity is caused, at least in part, by decreased Kv7.2/3 (KCNQ2/3) potassium currents. However, the biophysical mechanisms underlying resilience to tinnitus, which is observed in noise-exposed mice that do not develop tinnitus (non-tinnitus mice), remain unknown. Our results show that noise exposure induces, on average, a reduction in KCNQ2/3 channel activity in fusiform cells in noise-exposed mice by 4 days after exposure. Tinnitus is developed in mice that do not compensate for this reduction within the next 3 days. Resilience to tinnitus is developed in mice that show a re-emergence of KCNQ2/3 channel activity and a reduction in HCN channel activity. Our results highlight KCNQ2/3 and HCN channels as potential targets for designing novel therapeutics that may promote resilience to tinnitus.

## Introduction

Tinnitus, the perception of sound in the absence of acoustic stimulus, is frequently caused by exposure to loud sounds (*Shargorodsky et al., 2010*) and can be detrimental to the quality of life for millions of tinnitus sufferers (*Roberts et al., 2010*; *Shargorodsky et al., 2010*). Although the development of tinnitus is strongly correlated with acoustic trauma and noise-induced hearing loss, the absence of tinnitus after exposure to loud sounds—resilience to tinnitus—has been observed in a significant percentage of the population both in humans and in animal models (*Yankaskas, 2012*; *Zeng et al., 2012*; *Li et al., 2013*). Many studies have investigated the plasticity mechanisms underlying susceptibility to tinnitus; however, much less is known about the resilience mechanisms. Elucidating the resilience mechanisms could be vital for both developing pharmacological interventions to prevent and treat tinnitus, and understanding the mechanisms that differentiate pathogenic plasticity that leads to tinnitus from homeostatic plasticity that prevents the induction of tinnitus.

Here, we studied the resilience mechanisms in the dorsal cochlear nucleus (DCN), an auditory brainstem nucleus. Ablation of the DCN 3–5 months after the acoustic trauma did not affect the psychophysical evidence of tinnitus (*Brozoski and Bauer, 2005*). However, when bilateral DCN lesion occurred prior to noise exposure, it prevented the induction of tinnitus (*Brozoski et al., 2012*). These results suggest that although DCN is not essential for the maintenance of tinnitus, it is indispensable for the induction of tinnitus.

One of the major neural correlates for tinnitus induction is increased spontaneous firing rates in DCN fusiform cell, termed hyperactivity (*Zhang and Kaltenbach, 1998*; *Brozoski et al., 2002*; *Li et al., 2013*). Whereas a persistent change in neuronal firing properties in other auditory centers such as the ventral cochlear nucleus, inferior colliculus, auditory thalamus or auditory cortex may be

**\*For correspondence:** thanos@pitt.edu.

**Competing interests:** The authors declare that no competing interests exist.

**eLife digest** Tinnitus is often described as 'ringing in the ears'. Though the phantom sounds, which are heard in the absence of any genuine external noise, can take a variety of forms including buzzing, whistling, or humming. While training the brain to pay less attention to these internally generated sounds can sometimes reduce the impact of tinnitus, many people find that the disorder reduces their quality of life significantly.

One of the main causes of tinnitus is prolonged or repeated exposure to excessive noise. However, not everyone with such exposure develops tinnitus. Certain individuals appear to show a natural resilience. Identifying the basis of this resilience could make it possible to develop drugs that enhance these mechanisms, and thereby extend this protection to those who would otherwise be at risk of tinnitus.

Li et al. have brought this a step closer by studying tinnitus resilience mechanisms in mice. Previous work by the same group revealed that in mice with tinnitus a group of neurons in the brainstem called 'fusiform cells' are overly active. These cells receive direct input from the ear, and their hyperactivity is largely due to ion channels called KCNQ2/3 channels being less active. These channels allow for potassium ions to flow across the membrane and thereby control the activity of fusiform cells.

Li et al. now show that exposure to excessive noise causes a reduction in KCNQ2/3 activity in the exposed mice. However, in animals that successfully avoid developing tinnitus, KCNQ2/3 activity spontaneously recovers over the course of a few days. This recovery triggers a reduction in the activity of another type of ion channel, known as the HCN channel. The combined flexibility of KCNQ and HCN channels prevents tinnitus-associated hyperactivity in the fusiform cells.

Drugs that increase activity of KCNQ2/3 channels, and/or reduce activity of HCN channels, could thus boost resilience to tinnitus. In the future, targeting both channel types at the same time could provide an effective treatment with minimal side effects.

important for tinnitus maintenance (*Eggermont and Roberts, 2004*; *Vogler et al., 2011*; *Kalappa et al., 2014*; *Ropp et al., 2014*), transient increases in DCN neuronal spontaneous firing rates appear crucial for tinnitus induction (*Li et al., 2013*; *Ropp et al., 2014*).

After exposure to tinnitus-inducing sounds, key physiological properties of DCN fusiform cells are different between noise-exposed animals that develop tinnitus (tinnitus animals) and animals that do not develop tinnitus (non-tinnitus animals). Namely, although all noise-exposed animals show similar shifts in their hearing thresholds, only a portion of them display DCN hyperactivity and develop tinnitus (*Longenecker and Galazyuk, 2011*; *Koehler and Shore, 2013*; *Li et al., 2013*; *Longenecker et al., 2014*). Importantly, fusiform cell hyperactivity in tinnitus mice is associated with reduced KCNQ2/3 currents at hyperpolarized membrane potentials, due to a depolarizing shift in the voltage dependence of KCNQ2/3 channel opening (*Li et al., 2013*). This reduction, along with decreased synaptic inhibition and increased synaptic excitation, leads to DCN hyperactivity (*Wang et al., 2009*; *Middleton et al., 2011*; *Pilati et al., 2012*; *Zeng et al., 2012*). On the other hand, non-tinnitus mice do not show fusiform cell hyperactivity after sound exposure and express normal levels of KCNQ2/3 currents (*Li et al., 2013*). Nonetheless, knowledge of the biophysical mechanisms associated with resilience to tinnitus is still in its infancy.

Because in vitro studies have shown that the intrinsic properties of fusiform cells generate spontaneous firing in these neurons (*Leao et al., 2012*), we blocked excitatory and inhibitory synaptic transmission to investigate the intrinsic mechanisms underlying resilience to tinnitus in a mouse model of noise-induced tinnitus. We evaluated fusiform cell intrinsic excitability in control, tinnitus and non-tinnitus mice. We then extended this analysis by pharmacologically and biophysically isolating the ion channels whose noise-induced plasticity was associated with fusiform cell spontaneous firing rates and tinnitus or non-tinnitus behavior. Finally, we manipulated channel activity in vivo and tested the effects of these manipulations on the induction of tinnitus. Our results highlight the importance of KCNQ2/3 and HCN channels in the resilience to tinnitus and illuminate therapeutic paths that may enhance resilience for tinnitus prevention.

## Results

### Mouse model of tinnitus allows for behavioral separation of noise-exposed mice with either vulnerability or resilience to tinnitus

To study the neural mechanisms underlying resilience to noise-induced tinnitus, we employed an animal model of tinnitus that permits behavioral separation of tinnitus from non-tinnitus mice. According to this model, behavioral evidence of tinnitus is evaluated based on the inability of tinnitus mice to detect a silent sound gap in a continuous background sound, because their tinnitus 'fills in the gap' (*Turner et al., 2006*; *Longenecker and Galazyuk, 2011*; *Li et al., 2013*, but see *Hickox and Liberman, 2014* and 'Discussion'). When a silent gap is introduced in a constant background sound before a startle stimulus (gap trial), normal mice show reduced startle amplitude compared to their response to a startle stimulus preceded by the same background sound but without any gap (no-gap trial; *Figure 1A* top left, diagram showing gap and no-gap trials; *Figure 1A* top middle, gray and black, representative startle responses from a sham-exposed, control mouse). Noise-exposed mice that do not detect the silent gap show similar startle amplitudes in gap and no-gap trials and are considered tinnitus mice (*Figure 1A* top middle, gray and green, representative startle responses from a tinnitus mouse). Noise-exposed mice that detect the silent gap show reduced startle amplitudes in gap trials and are considered non-tinnitus mice (*Figure 1A* top right, gray and blue, representative startle responses from a non-tinnitus mouse). To quantify the gap detection ability of control and noise-exposed mice, we calculated the gap startle ratio before and after exposure (*Figure 1B*). Gap startle ratio is the maximum startle amplitude in gap trials divided by the maximum startle amplitude in no-gap trials. Moreover, to investigate the frequency specificity of noise-induced tinnitus, we quantified the gap startle ratio for background sounds with different frequencies ('Materials and methods'). Our results showed that 7 days after noise exposure, 52.4% (11/21) of the noise-exposed mice exhibited a significant gap detection deficit at high-frequency (20–32 kHz) but not at low-frequency (10–16 kHz) background sounds. This deficit is indicated by an increase in gap startle ratios and is consistent with behavioral evidence of high-frequency tinnitus (*Figure 1B* middle; *Figure 1—figure supplement 1A–D*; 'Materials and methods').

Increases in gap startle ratios in tinnitus mice are not consistent with noise-induced impairments in temporal processing or noise-induced inability to hear the background sound, because prepulse inhibition, PPI, was similar among control, tinnitus, and non-tinnitus mice (*Figure 1A* bottom left, diagram showing startle-only and prepulse trials; *Figure 1A* bottom middle and right, representative startle responses from control, tinnitus and non-tinnitus mice; *Figure 1C*, summary graphs of PPI ratio; *Figure 1—figure supplement 1E–G*, 'Materials and methods'). PPI ratio reflects the inhibition of startle response by a preceding non-startling sound of similar intensity as the background sound used in gap detection (*Figure 1C*). Moreover, tinnitus and non-tinnitus mice displayed similar hearing thresholds before and after noise-exposure, as evidenced by their similar auditory brainstem response (ABR) thresholds (*Figure 2A,B*; 'Materials and methods'). In response to acoustic stimuli, ABRs reflect the synchronous activity of auditory brainstem nuclei arising from the auditory nerve (wave I) to the inferior colliculus, IC (wave V, *Figure 2A*). Similar ABR thresholds may be accompanied with hidden differences in suprathreshold amplitudes of wave I. Because these differences could reflect differential degeneration of the auditory nerve (*Kujawa and Liberman, 2009*; *Furman et al., 2013*), we compared the wave I amplitude of ABRs in response to suprathreshold sounds between tinnitus and non-tinnitus mice, before and after noise exposure. Whereas wave I amplitudes were reduced after noise-exposure for all noise-exposed mice, tinnitus and non-tinnitus mice showed no differences in wave I amplitude, suggesting no difference in the damage of afferent nerve terminals between these two groups (*Figure 2C,D*). Taken together, our results suggest that neither differential noise-induced hearing threshold shifts nor differential auditory nerve damage can explain the behavioral differences between tinnitus and non-tinnitus mice.

### Bidirectional plasticity of KCNQ2/3 channels in the DCN is crucial for vulnerability and resilience to tinnitus

Next we explored whether behavioral differences between tinnitus and non-tinnitus mice are due to differential intrinsic plasticity in DCN fusiform cells, which are the first targets of the auditory nerve in the central nervous system. Tinnitus mice present reduced KCNQ2/3 channel activity and increased

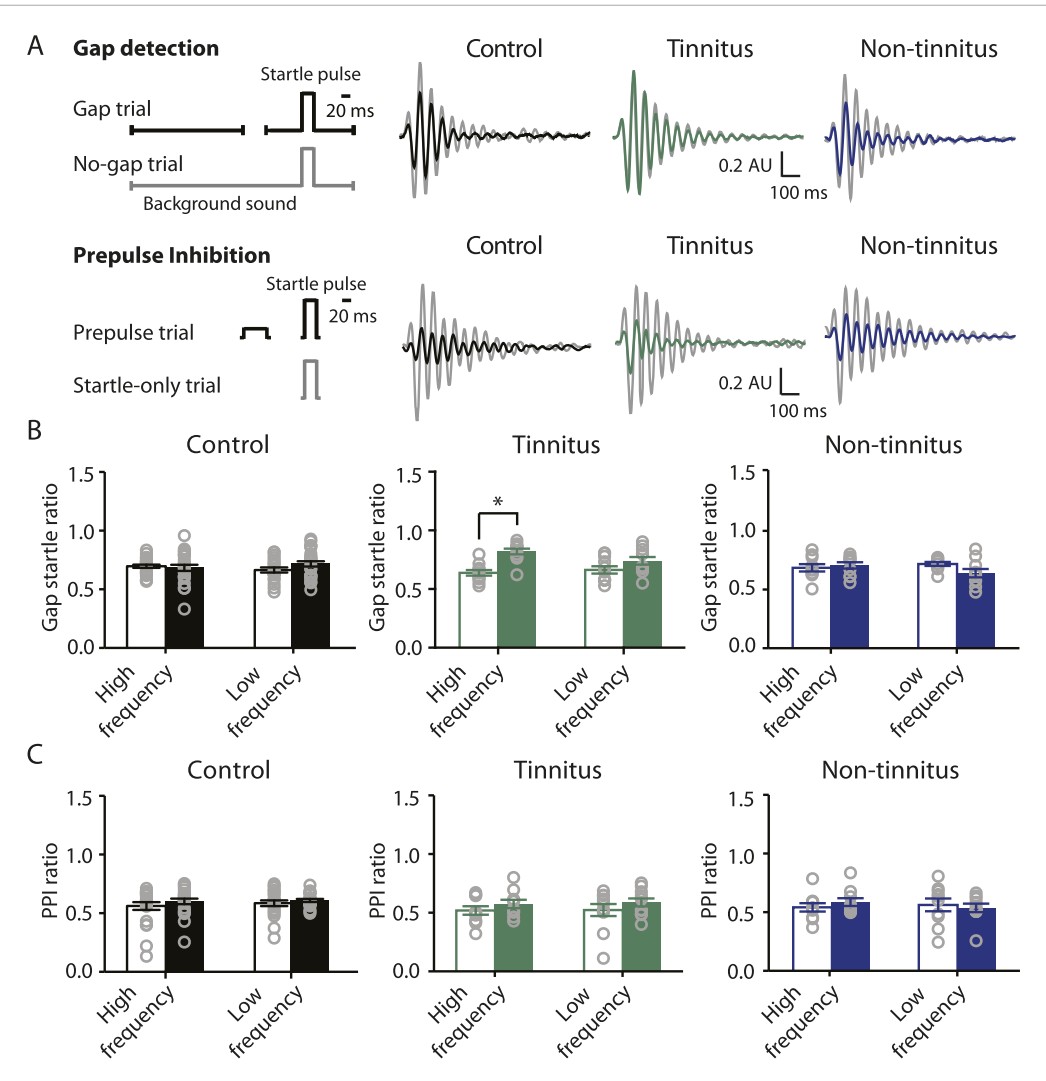

**Figure 1**. Mouse model of tinnitus allows for behavioral separation of noise-exposed mice with either vulnerability or resilience to tinnitus. **A**. Top left: diagram illustrating gap and no-gap trials in the gap detection behavioral assay. Top right: representative startle responses in no-gap (always in gray) and gap trials from control (black), tinnitus (green), and non-tinnitus (blue) mice. AU: Arbitrary unit. Bottom left: diagram illustrating startle-only and prepulse trials in the prepulse inhibition (PPI) behavioral assay. Bottom right: representative startle responses in startle-only (always in gray) and prepulse trials from control (black), tinnitus (green), and non-tinnitus (blue) mice. **B**. Summary graphs of gap startle ratio (response to gap trial/response to no-gap trial) before (open bar) and 1 week after noise exposure (filled bar) for high- and low-frequency background sounds (high-frequency background, 20–32 kHz, control: n = 21, p = 0.6; tinnitus: n = 11, p < 0.001; non-tinnitus: n = 10, p = 0.6; low frequency background, 10–16 kHz, control: n = 21, p = 0.42; tinnitus: n = 11, p = 0.06; non-tinnitus: n = 10, p = 0.07). **C**. Summary graphs of prepulse inhibition ratio (PPI ratio, response to prepulse trial/response to startle-only trial) before (open bar) and 1 week after noise exposure (filled bar) for high- and low-frequency prepulse (high-frequency background, 20–32 kHz, control: n = 21, p = 0.25; tinnitus: n = 11, p = 0.18; non-tinnitus: n = 10, p = 0.36; low frequency background, 10–16 kHz, control: n = 22, p = 0.56; tinnitus: n = 11, p = 0.17; non-tinnitus: n = 10, p = 0.56). Asterisk, p < 0.05. Error bars indicate SEM. See end of the manuscript for detailed values in **B** and **C**.

The following figure supplement is available for figure 1:

**Figure supplement 1**. Tinnitus behavior (gap detection deficit) is detected with high-frequency background sounds.

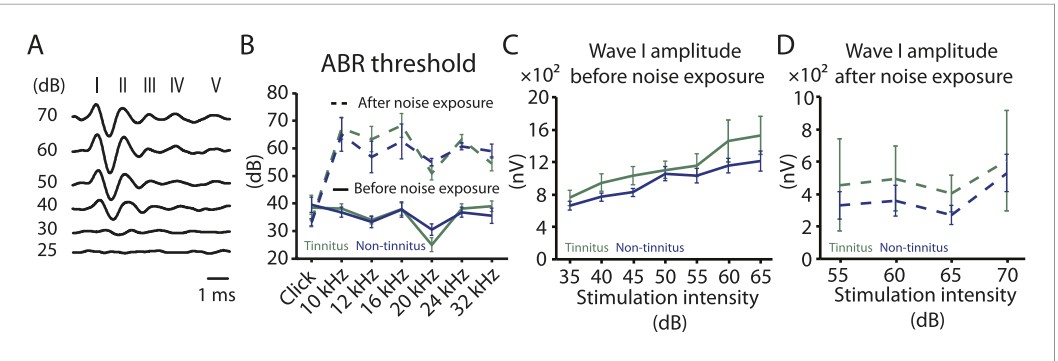

**Figure 2**. Tinnitus and non-tinnitus mice show similar ABR thresholds and suprathreshold ABR wave I amplitudes. **A**. Representative raw traces of auditory brainstem response (ABR) in response to click tone presented at different intensities (dB). I–V represent the different waves of the ABR. **B**. Summary graph of ABR thresholds for tinnitus (green) and non-tinnitus (blue) mice before (solid line) and 7 days (dashed line) after noise exposure (n = 5–11, no statistical difference was observed between tinnitus and non-tinnitus mice). **C**. Summary graph of suprathreshold wave I amplitudes for tinnitus (green) and non-tinnitus (blue) mice before noise exposure for high frequency (20–32 kHz) sound stimulation (n = 12–25, no statistical difference was observed between tinnitus and non-tinnitus mice). **D**. Summary graph of suprathreshold wave I amplitudes for tinnitus (green) and non-tinnitus (blue) mice 7 days after noise exposure for high frequency (20–32 kHz) sound stimulation (n = 4–10, no statistical difference was observed between tinnitus and non-tinnitus mice). See end of the manuscript for detailed values for **B**–**D**. Error bars indicate SEM.

spontaneous firing rates in DCN fusiform cells 7 days after noise exposure (*Li et al., 2013*). In contrast, non-tinnitus mice are associated with normal KCNQ2/3 channel activity and normal spontaneous firing rates of fusiform cell (*Li et al., 2013*). These findings suggest that in resilient, non-tinnitus mice either there is no transient reduction in KCNQ2/3 channel activity post noise exposure or that KCNQ2/3 channel activity is transiently reduced but recovers by 7 days post noise exposure. To distinguish between these two possibilities, we measured KCNQ2/3 current amplitudes in fusiform cells from noise-exposed mice 4 days post noise exposure. Because fusiform cells from DCN regions that represent high, but not low, frequency sounds are involved in tinnitus-related biophysical changes (*Li et al., 2013*), all in vitro electrophysiological recordings were conducted on fusiform cells from high-frequency DCN regions (≥20 kHz, dorsal part). To quantify KCNQ2/3 currents, we held fusiform cells at −30 mV for 5 s and then stepped the voltage to −50 mV for 1 s to unmask the slow deactivation of KCNQ2/3 channels (*Figure 3A*). Consistent with previous measurements of KCNQ2/3 currents in fusiform cells (*Li et al., 2013*), this protocol revealed a slowly deactivating current, which was significantly reduced by XE991 application (10 μM, a specific KCNQ channel blocker; *Figure 3A*). Because KCNQ2/3 heteromeric channels mediate the KCNQ currents in fusiform cells (*Li et al., 2013*), the XE991-sensitive component of these recordings represents the KCNQ2/3 current amplitude. 4 days noise-exposed mice showed a significant reduction in KCNQ2/3 current amplitude (*Figure 3B*; 'Materials and methods'). A Boltzmann fit of the KCNQ2/3 conductance–voltage (G–V) function showed that the $G_{max}$ of KCNQ2/3 currents was not different between 4 days sham- and 4 days noise-exposed mice (*Figure 3C,E*), but the $V_{1/2}$ was shifted to more depolarized potentials in the 4 days noise-exposed mice (*Figure 3C,D*; 'Materials and methods'). These results suggest that the reduction of KCNQ2/3 currents in 4 days noise-exposed mice is due, at least in part, to a depolarizing shift in the $V_{1/2}$ of KCNQ2/3 channels, which is mechanistically similar to the reduction of KCNQ2/3 currents in tinnitus mice when assessed 7 days after noise exposure (*Li et al., 2013*).

Given that fusiform cells from non-tinnitus mice display control-level KCNQ2/3 currents 7 days post noise exposure (*Li et al., 2013*), our results suggest that it is the recovery in KCNQ2/3 channel activity, not the lack of reduction in KCNQ2/3 currents, which is linked with the resilience to tinnitus. Moreover, previous findings showed that in vivo pharmacological activation of KCNQ currents with either KCNQ2-5 or KCNQ2/3 channel activators, retigabine or SF0034, respectively

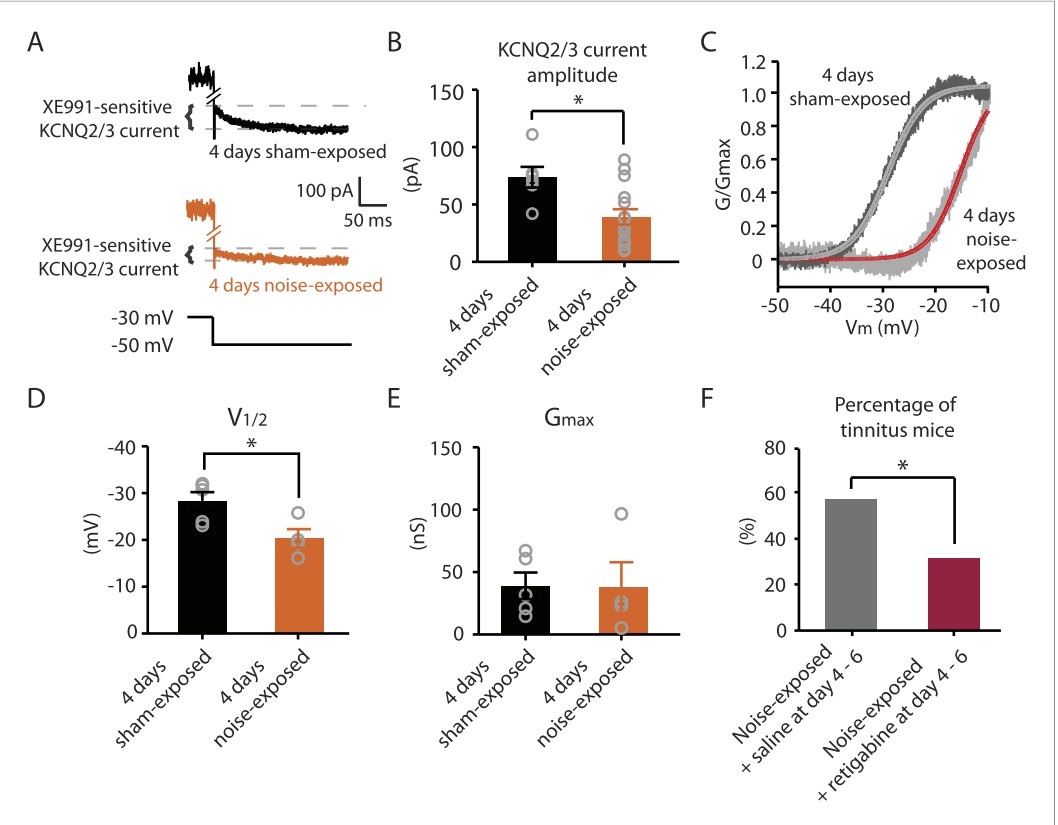

**Figure 3**. Noise-exposed mice show reduced KCNQ2/3 channel activity 4 days after noise exposure; this reduction is due to a depolarizing shift of $V_{1/2}$. **A**. Representative traces illustrating XE991 (10 µM)-sensitive KCNQ2/3 currents (Top) in fusiform cells from a 4 days sham-exposed (black) and a 4 days noise-exposed (yellow) mouse in response to a voltage step to −50 mV from a holding potential of −30 mV (Bottom; for clearer representation, currents were truncated along time axis). **B**. Summary graph showing KCNQ2/3 current amplitude in fusiform cells from 4 days sham-exposed and 4 days noise-exposed mice (4 days sham-exposed mice, $73.8 \pm 9.05$ pA, n = 6; 4 days noise-exposed mice, $39.16 \pm 6.7$ pA, n = 15, p = 0.01). **C**. Representative conductance–voltage relationship of XE991-sensitive current in 4 days sham-exposed (dark gray) and 4 days noise-exposed mice (light gray). Gray and red lines represent Boltzmann fits. **D**. Summary graph for Boltzmann fit parameter $V_{1/2}$ (4 days sham-exposed: $−28.3 \pm 1.9$ mV, n = 5, 4 days noise-exposed: $−20.3 \pm 2.0$ mV, n = 4, p = 0.03). **E**. Summary graph for Boltzmann fit parameter $G_{max}$ (4 days sham-exposed: $39.1 \pm 10.2$ nS, n = 5, 4 days noise-exposed: $37.7 \pm 20.2$ nS, n = 4, p = 0.90). **F**. Effect of retigabine injection 4–6 days after noise exposure on the percentage of mice that develop tinnitus. (Noise-exposed mice + saline at day 4–6: 57.1%, n = 14, noise-exposed mice + retigabine at day 4–6: 31.3%, n = 16, p = 0.03). Asterisk, p < 0.05. Error bars indicate SEM.

(*Tatulian et al., 2001*; *Kalappa et al., 2015*), prevented the development of tinnitus (*Li et al., 2013*; *Kalappa et al., 2015*). The ability of KCNQ channel activators to prevent the development of tinnitus is probably occurring through the reversal of the pathogenic reduction of KCNQ2/3 channel activity and the promotion of the natural resilience to tinnitus. However, in those experiments KCNQ channel activators were administered 30 min after noise exposure and then twice a day for an additional 5 days. Here, we tested whether retigabine application initiated at day 4, instead of 30 min after noise exposure, is sufficient to reduce the incidence of tinnitus at day 7. Application of retigabine at day 4 significantly reduced the percentage of mice that developed tinnitus (*Figure 3F*; 'Materials and methods'), suggesting that the recovery of KCNQ2/3 currents between day 4 and 7 is crucial for tinnitus resilience. Taken together, our results suggest that experience-dependent bidirectional plasticity of KCNQ2/3 channel activity is associated with the vulnerability and resilience to noise-induced tinnitus: experience-dependent depression in KCNQ2/3 channel activity is important for vulnerability, while experience-dependent recovery in KCNQ2/3 channel activity is important for resilience to tinnitus.

## Reduced HCN channel activity in fusiform cells from non-tinnitus mice underlies biophysical differences between control and non-tinnitus mice

Our results highlight that pathogenic (reduction) and homeostatic plasticity (recovery) in KCNQ2/3 channel activity are associated with tinnitus and non-tinnitus behavior, respectively. However, it is not known whether fusiform cells from non-tinnitus mice are biophysically similar to the fusiform cells from control mice, or they have undergone homeostatic plasticity in other channels, which could further contribute to tinnitus resilience. To answer this question, we compared the biophysical properties between fusiform cells from control and non-tinnitus mice 7 days after sham or noise exposure. We found that although control mice and non-tinnitus mice show similar non-tinnitus behavior (*Figure 1B*), similar fusiform cell spontaneous firing rates (*Li et al., 2013*), similar levels of KCNQ2/3 channel activity (*Li et al., 2013*), and similar spike parameters (*Supplementary file 1*; 'Materials and methods'), non-tinnitus mice are biophysically distinct from control mice. When we blocked fusiform cell spontaneous firing activity with tetrodotoxin (0.5 µM, a specific sodium channel blocker), we found that fusiform cells from non-tinnitus mice showed more hyperpolarized resting membrane potential (RMP; *Figure 4A*). Because fusiform cell RMP is similar between control and tinnitus mice (*Li et al., 2013*), this finding suggests specific changes in subthreshold ionic conductance(s) occurring only in non-tinnitus mice. Moreover, when we used small current steps to evaluate the onset and steady state input resistance ($R_{in}$; *Figure 4D*; 'Materials and methods'), we found that onset $R_{in}$ was not different among control, tinnitus, and non-tinnitus mice (*Figure 4B*). However, non-tinnitus mice displayed a significantly increased steady-state $R_{in}$, which was revealed later in the voltage response (*Figure 4C*; 'Materials and methods'). Together, our

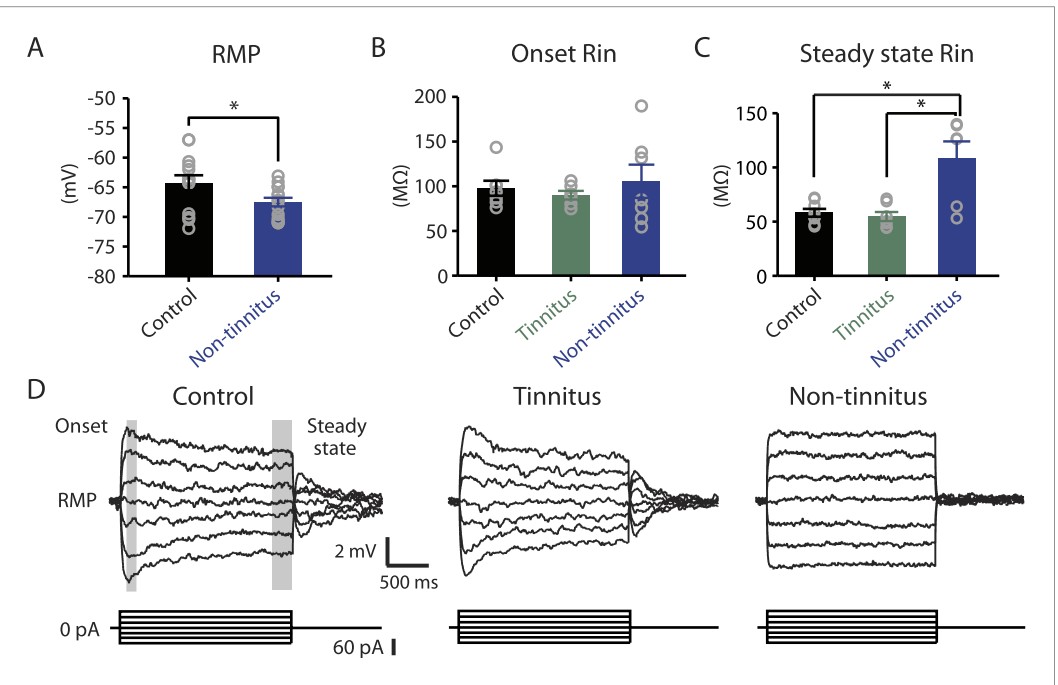

**Figure 4**. Non-tinnitus mice are biophysically distinct from control mice. **A**. Summary graph of resting membrane potential (RMP) of fusiform cells from control and non-tinnitus mice after blocking spontaneous firing with 0.5 µM TTX (control: −64.2 ± 1.3 mV, n = 14; non-tinnitus: −67.5 ± 0.7 mV, n = 15, p = 0.04). **B**. Summary graph of onset input resistance ($R_{in}$) of fusiform cells as measured in **D** from control, tinnitus and non-tinnitus mice (control: 97.9 ± 8.4 MΩ, n = 8; tinnitus: 90.1 ± 4.8 MΩ, n = 7; non-tinnitus: 105.7 ± 18.5 MΩ, n = 7, p = 0.73). **C**. Summary graph of steady-state $R_{in}$ of fusiform cells as measured in **D** from control, tinnitus, and non-tinnitus mice (control: 58.2 ± 3.6 MΩ, n = 8; tinnitus: 54.8 ± 4.2 MΩ, n = 7; non-tinnitus: 108.18 ± 15.9 MΩ, n = 7, p = 0.04). **D**. Representative voltage traces of fusiform cells from control, tinnitus, and non-tinnitus mice (Top) in response to small depolarizing and hyperpolarizing current steps (Bottom, −60 pA–60 pA, 20 pA step) for measuring input resistance. Shaded areas indicate the region for evaluating onset (starting from the peak of the voltage response, 100 ms width) and steady-state $R_{in}$ (starting at the voltage response 1.75 s after the current injection, 250 ms width). Asterisk, p < 0.05. Error bars indicate SEM.

results revealed that fusiform cells from non-tinnitus mice exhibit a decrease in a slowly activating and deactivating depolarizing conductance that is open at subthreshold potentials. These results indicate that although control and non-tinnitus mice show similar non-tinnitus behavior, non-tinnitus mice are biophysically distinct from control mice. Importantly, these results highlight the involvement of homeostatic plasticity of additional, non-KCNQ, ionic conductances in underlying resilience to tinnitus.

Hyperpolarization-activated cyclic nucleotide-gated channels (HCN channels) exhibit slowly activating and deactivating kinetics and are important regulators of subthreshold dynamics in fusiform cells (*Leao et al., 2012*). We therefore hypothesized that a decrease in HCN channel activity mediates the hyperpolarized RMP and the increased steady-state $R_{in}$ in non-tinnitus mice. To test this hypothesis, we injected hyperpolarizing current to activate HCN channels. HCN channel activation led to a characteristic rebound in the membrane voltage (voltage sag) that was sensitive to the specific HCN channel blocker ZD7288 (10 μM; *Figure 5A*, red: before ZD7288; black: after ZD7288). We quantified HCN channel activity by measuring the ZD7288-sensitive sag ratio, which is the difference between the peak voltage response ($V_{peak}$) and steady-state voltage response ($V_{ss}$), normalized to the $V_{peak}$ (Sag ratio = ($V_{peak}$–$V_{ss}$)/$V_{peak}$*100%). Consistent with our hypothesis, fusiform cells from non-tinnitus mice showed significantly reduced HCN channel activity (*Figure 5B*). To confirm whether the reduction in HCN channel activity was responsible for the non-tinnitus-specific intrinsic properties, we evaluated the effect of ZD7288 on steady-state $R_{in}$ and RMP. Indeed, ZD7288 application abolished the differences in steady-state input resistance and in RMP among control, tinnitus, and non-tinnitus mice (*Figure 5C–E*). Together, our results are consistent with the

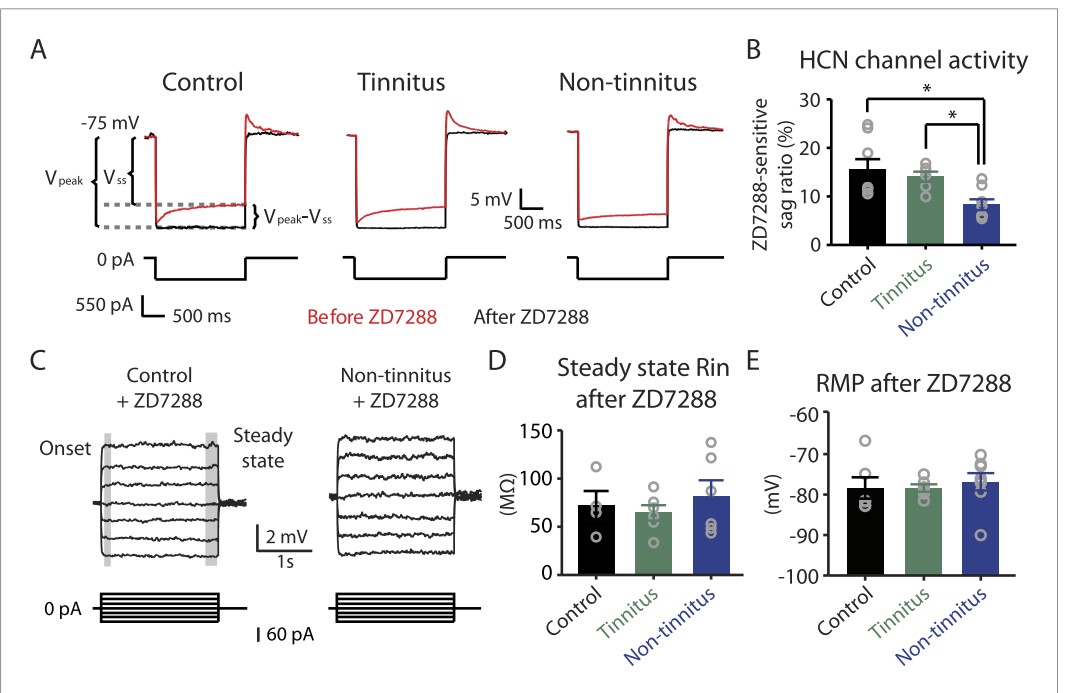

**Figure 5**. Reduced HCN channel activity in non-tinnitus mice underlies the biophysical differences between control and non-tinnitus mice. **A**. Representative voltage traces (Top) of fusiform cells from control, tinnitus, and non-tinnitus mice in response to a hyperpolarizing current step (Bottom) for measuring HCN channel activity before (red) and after (black) 10 μM ZD7288. HCN channel activity is measured by calculating the sag ratio ($V_{peak}$–$V_{ss}$)/$V_{peak}$*100 (%) that is sensitive to ZD7288. **B**. Summary graph showing HCN channel activity as measured by the protocol described in **A** (control: 15.5 ± 2.2%, n = 8; tinnitus: 14.1 ± 1.1%, n = 6; non-tinnitus: 8.3 ± 1.1%, n = 8, p = 0.02). **C**. Representative voltage traces of fusiform cells from control and non-tinnitus mice in response to current steps for measuring $R_{in}$ as in *Figure 4D* but now in the presence of 10 μM ZD7288. **D**. Summary graph showing steady-state $R_{in}$ in control, tinnitus, and non-tinnitus mice in 10 μM ZD7288 (control, 72.0 ± 15.1 MΩ, n = 4; tinnitus, 64.4 ± 8.0 MΩ, n = 6; non-tinnitus, 81.9 ± 16.4 MΩ, n = 6, p = 0.9). **E**. Summary graph showing resting membrane potential (RMP) in control, tinnitus, and non-tinnitus mice in 10 μM ZD7288 (control, −78.4 ± 2.6 mV, n = 6; tinnitus, −78.4 ± 0.9 mV, n = 6; non-tinnitus, −76.8 ± 2.1 mV, n = 8, p = 0.4). Asterisk, p < 0.05. Error bars indicate SEM.

notion that reduced HCN channel activity is a major contributor for the more hyperpolarized RMP and increased steady-state $R_{in}$ in non-tinnitus mice. Moreover, our results show that noise-induced reduction in HCN channel activity is another critical biophysical change associated with resilience to tinnitus.

## Pharmacological activation of KCNQ channels prevents the development of tinnitus and promotes the reduction in HCN channel activity

Because both recovery in KCNQ currents and reduction in HCN channel activity are associated with resilience to tinnitus, we hypothesized that either KCNQ plasticity occurs before HCN channel plasticity and drives the resilience pathway or that plasticity in HCN channels occurs before KCNQ channel plasticity and plays the determinant role for tinnitus resilience. To distinguish between these two possibilities, we measured HCN channel activity 4 days after noise exposure. Our results revealed that although noise-exposed mice showed significantly reduced KCNQ2/3 currents in fusiform cells 4 days after noise exposure (*Figure 3B*), HCN channel activity was not different between sham-exposed and noise-exposed mice (*Figure 6A,B*). These results indicate that the decrease in KCNQ2/3 activity happens before the reduction of HCN channel activity.

Given the initial appearance of KCNQ current reduction 4 days after noise exposure and because pharmacological activation of KCNQ currents with retigabine 4 days after noise exposure is sufficient to prevent the development of tinnitus (*Figure 3F*), we hypothesized that the recovery of KCNQ channels not only drives resilience to tinnitus but also promotes the reduction in HCN channel activity in non-tinnitus mice. To test this hypothesis, we measured the effect of retigabine on HCN current amplitude (intraperitoneal [IP] injections, 10 mg/kg 30 min post exposure then twice daily for 5 days; 'Materials and methods'). Consistent with our hypothesis, we found that fusiform cells from retigabine-injected mice, which showed reduced incidence of tinnitus and reduced spontaneous firing rate of fusiform cells (*Figure 7A*, *Figure 7—figure supplement 1A,C,D*), displayed reduced HCN channel activity (*Figure 6C,D*). For some mice, we used flupirtine, another KCNQ channel activator (*Mackie and Byron, 2008*; 'Materials and methods'). Moreover, application of KCNQ channel activators did not affect ABR thresholds, wave I amplitude, or PPI ratios (*Figure 7B–D*, *Figure 7—figure supplement 1B*), suggesting that these activators do not affect either noise-induced hearing threshold shifts or auditory nerve damage. These results are consistent with

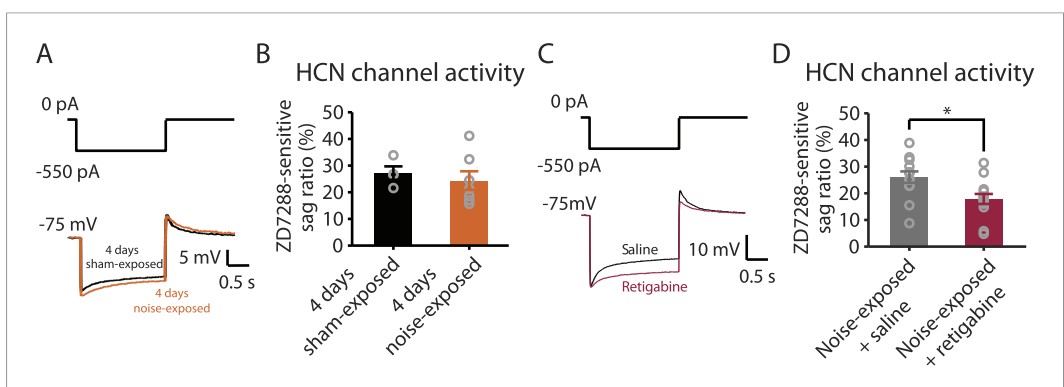

**Figure 6**. Noise-induced HCN plasticity occurs after KCNQ2/3 plasticity; KCNQ2/3 enhancement promotes HCN channel activity reduction. **A**. Representative voltage traces (Bottom) from fusiform cells in response to hyperpolarizing current step (Top) for measuring HCN channel activity in 4 days sham-exposed (black) and 4 days noise-exposed mice (yellow). **B**. Summary graph showing HCN channel activity in fusiform cells from 4 days sham-exposed and 4 days noise-exposed mice, measured as in *Figure 5A,B* (4 days sham-exposed mice, 27.2 ± 2.5%, n = 4; 4 days noise-exposed mice, 24.4 ± 3.5%, n = 7, p = 0.7). **C**. Representative voltage traces (Bottom) from fusiform cells in response to hyperpolarizing current step (top) for measuring HCN channel activity in noise-exposed with saline injection (black) and noise-exposed mice with retigabine injection (red). **D**. Summary graph showing HCN channel activity, as measured as in *Figure 5A,B*. (Noise-exposed and saline-injected mice, 26.1 ± 0.6%, n = 13; noise-exposed retigabine-injected, 17.9 ± 0.5%, n = 14, p = 0.008). Asterisk, p < 0.05. Error bars indicate SEM.

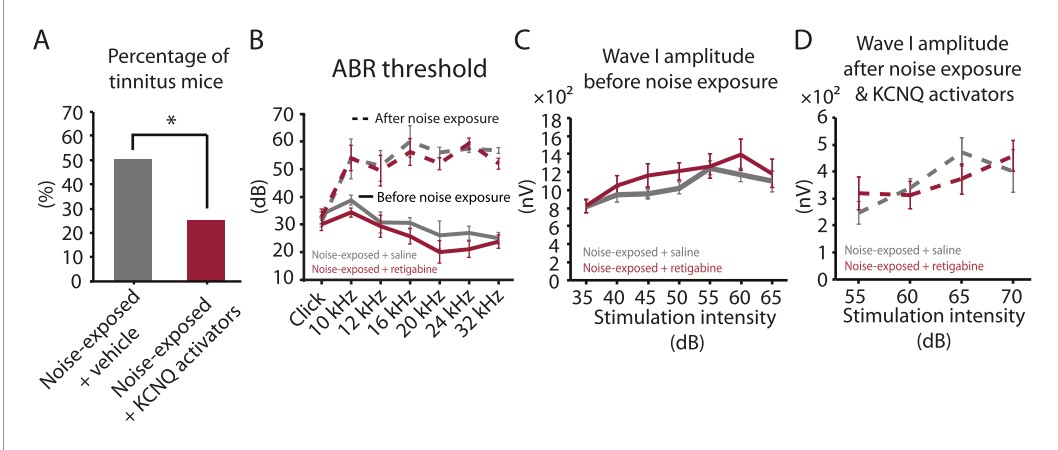

**Figure 7**. Injection of KCNQ activators after noise exposure reduces the incidence of tinnitus development without affecting threshold and suprathreshold ABRs. **A**. Percentage of mice that develop tinnitus (noise-exposed mice with intraperitoneal (IP) injection of vehicle, 50%, n = 18, noise-exposed mice with IP injection KCNQ channel activators, 25%, n = 20, p = 0.02). For noise-exposed mice with IP injection of vehicle (11 for retigabine vehicle and 7 flupirtine vehicle at 10 mg/kg); for noise-exposed mice with IP injection of KCNQ channel activators (10 for retigabine and 10 for flupirtine at 10 mg/kg). **B**. Summary graph of ABR thresholds from saline- (gray) and retigabine-injected (red) mice before (solid line) and 7 days after (dashed line) noise exposure and injection (n = 4–9, no statistical difference was observed between retigabine- and saline-injected mice). **C**. Summary graph of suprathreshold wave I amplitudes for noise-exposed mice + saline (gray) and noise-exposed mice + retigabine (red) before noise exposure for high frequency (20–32 kHz) sound stimulation (n = 5–15, no statistical difference was observed between retigabine- and saline-injected mice). **D**. Summary graph of suprathreshold wave I amplitudes for noise-exposed mice + saline (gray) and noise-exposed mice + retigabine (red) after noise exposure and injection for high-frequency (20–32 kHz) sound stimulation (n = 5–20, no statistical difference was observed between retigabine- and saline-injected mice). See end of the manuscript for detailed values for **B**–**D**. Asterisk, p < 0.05. Error bars indicate SEM.

The following figure supplement is available for figure 7:

**Figure supplement 1**. In vivo administration of KCNQ channel activators prevents the development of tinnitus and reduces the spontaneous firing rate of fusiform cells.

recent studies showing that application of retigabine does not have any effect on hearing thresholds (*Sheppard et al., 2015*). However, application of retigabine reduced hearing loss in an animal model of sodium salicylate-induced sensorineural hearing loss and tinnitus (*Sheppard et al., 2015*). This is probably due to the fact that different mechanisms mediate sodium salicylate- and noise-induced hearing loss. Taken together, our results suggest that increases in KCNQ2/3 channel activity promote a decrease in fusiform cell HCN channel activity and resilience to tinnitus.

## Biophysical changes that lead to hyperactivity are associated with the induction of tinnitus

4 days after noise exposure, fusiform cells from noise-exposed mice displayed reduced KCNQ2/3 currents and control level HCN channel activity. This biophysical profile is similar to the fusiform cell profile from tinnitus mice 7 days post exposure. Therefore, we hypothesized that 4 days noise-exposed mice would show behavioral evidence of tinnitus and fusiform cell hyperactivity. Surprisingly, 4 days noise-exposed mice showed normal fusiform cell spontaneous firing activity and no behavioral evidence of tinnitus (*Figure 8A,B*, *Figure 8—figure supplement 1*). The lack of association between reduced KCNQ2/3 channel activity and fusiform cell hyperactivity may be explained by the hyperpolarized RMP in fusiform cells from 4 days noise-exposed mice (*Figure 8C*), which is probably due to plasticity in some other ionic conductance(s) ('Discussion'). These results suggest that biophysical changes that are associated with increases in fusiform cell spontaneous firing rates lead to tinnitus, while biophysical changes that maintain normal levels of spontaneous firing rates are associated with tinnitus resilience.

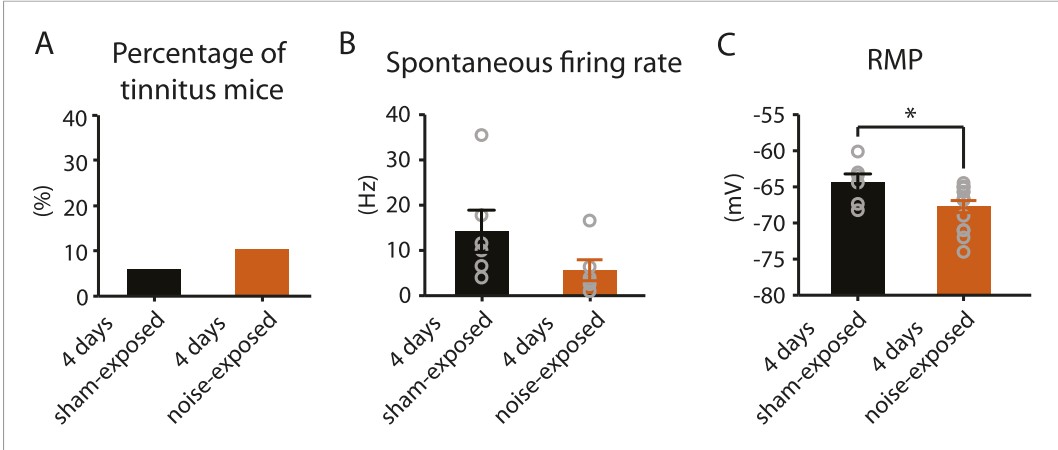

**Figure 8**. 4 days after noise exposure, mice have reduced KCNQ2/3 current amplitude but do not show either hyperactivity or tinnitus. **A**. Percentage of mice that develop tinnitus in 4 days sham-exposed and 4 days noise-exposed mice (4 days sham-exposed: n = 19; 4 days noise-exposed: n = 20, p = 0.28). **B**. Summary graph showing spontaneous firing rate in fusiform cells, assessed with whole-cell, voltage-follower mode recordings (current clamp, at I = 0), from 4 days sham-exposed and 4 days noise-exposed mice (4 days sham-exposed: 14.3 ± 4.7 Hz, n = 6; 4 days noise-exposed: 5.6 ± 2.3 Hz, n = 6, p = 0.13). **C**. Summary graph showing resting membrane potential (RMP) in 4 days sham-exposed mice and 4 days noise-exposed mice (4 days sham-exposed: −64.4 ± 1.2 mV, n = 6; 4 days noise-exposed: −67.7 ± 0.8 mV, n = 14, p = 0.04). Asterisk, p < 0.05. Error bars indicate SEM.

The following figure supplement is available for figure 8:

**Figure supplement 1**. 4 days noise-exposed mice exhibit similar gap detection and PPI compared to 4 days sham-exposed mice.

Taken together, we show that noise exposure leads to down regulation of KCNQ2/3 channel activity by 4 days after noise exposure. At this time, no tinnitus has developed yet, probably due to the absence of fusiform cell hyperactivity. Mice that show a natural recovery of KCNQ2/3 channel activity and a reduction in HCN channel activity display normal level of spontaneous firing rates and are resilient to tinnitus. Mice that show preservation of reduced KCNQ2/3 channel activity until 7 days post noise exposure show fusiform cell hyperactivity and develop tinnitus (**Figure 9**).

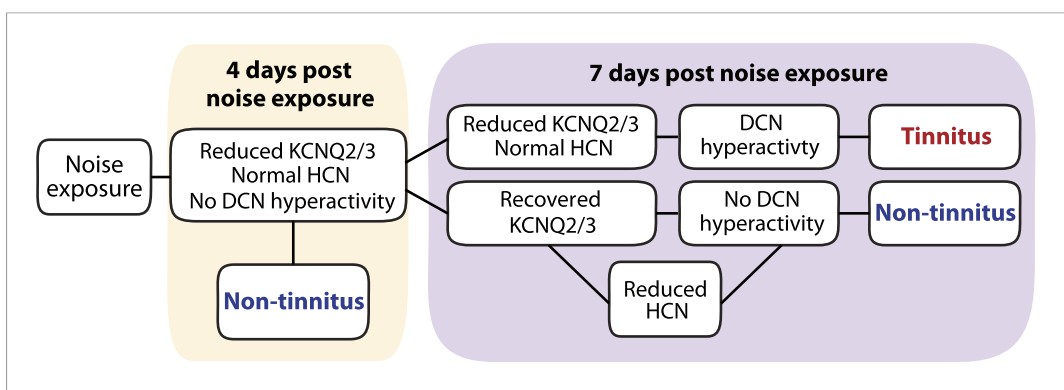

**Figure 9**. Biophysical mechanisms underlying the development of vulnerability and resilience to noise-induced tinnitus. Diagram illustrating the noise-induced plasticity of KCNQ2/3 and HCN channel activity, the emergence of DCN hyperactivity, and the development of vulnerability and resilience to tinnitus.

## Discussion

### Gap detection in tinnitus animal models and in humans with tinnitus

Given the subjective nature of tinnitus perception, there is not currently any behavioral paradigm in either animals or humans that provides for objective evidence of tinnitus. Despite this inherent limitation for studying mechanisms underlying tinnitus perception, deficits in the gap-prepulse inhibition of the acoustic startle reflex (gap detection) is the most widely used behavioral test for assessing behavioral evidence of tinnitus in animals (*Galazyuk and Hebert, 2015*). Research based on this behavioral assay has revealed important findings for the mechanisms underlying tinnitus, including neuronal hyperactivity, increased neuronal bursting activity, increased neuronal synchrony, degradation of frequency tonotopy, reduced GABAergic/glycinergic inhibition, enhanced GABAergic tonic inhibition, enhanced glutamatergic excitation, and changes in the biophysical properties of KCNQ channels (*Engineer et al., 2011*; *Middleton et al., 2011*; *Wang et al., 2011*; *Zeng et al., 2012*; *Li et al., 2013*; *Kalappa et al., 2014, 2015*; *Sametsky et al., 2015*). In several cases, these findings have led to ongoing clinical trials. For example, a recent study on the role of vagus nerve stimulation paired with tones for treating tinnitus was based on the gap detection paradigm in rats (*Engineer et al., 2011*) and is currently under clinical trial. Our studies, which also utilized the gap detection paradigm, revealed a novel KCNQ2/3 channel activator as a clinical candidate for preventing tinnitus (*Kalappa et al., 2015*).

Because the gap detection paradigm is based on a reflex, unlike other animal models of tinnitus requiring extensive behavioral training (*Bauer and Brozoski, 2001*; *Lobarinas et al., 2004*), it allows for faster tinnitus screening of a larger number of animals and permits the separation of tinnitus from non-tinnitus animals. However, the interpretation that tinnitus 'fills in' the gap in the background sound that precedes the startle stimulus is not compatible with a recent study, which showed that gap detection deficits in noise-exposed mice were evident only when the gap was placed immediately, but not 80 ms, before the startle stimulus (*Hickox and Liberman, 2014*). If tinnitus just 'fills in' the gap, then gap detection deficits in tinnitus animals are expected to be independent of where the silent gap is placed. Moreover, although human studies revealed gap detection deficits in tinnitus sufferers, which are consistent with the use of the gap detection in animal studies, the frequency of the tinnitus percept did not fully match the frequency of the background sound where gap detection deficits were observed (*Fournier and Hebert, 2013*). This lack of correspondence in frequency is also inconsistent with the 'fill in' theory. Other human studies, which used higher-level cognitive tasks but not the acoustic reflex for assessing gap detection, did not reveal gap detection deficits in tinnitus sufferers (*Campolo et al., 2013*; *Boyen et al., 2015*). Taken together, we propose that although caution should be taken when interpreting gap detection deficits, the gap-prepulse inhibition of the acoustic startle reflex is a useful behavioral assay for elucidating tinnitus mechanisms.

### Plasticity of KCNQ2/3 channels in hyperexcitability-related brain disorders

KCNQ2/3 channels are slowly activating, non-inactivating voltage-dependent potassium channels that are open at hyperpolarized (subthreshold) voltages. As a result, they control subthreshold membrane potential and serve as a powerful brake of neuronal firing activity (*Maljevic et al., 2008*; *Brown and Passmore, 2009*). Disorders that are characterized by neuronal hyperexcitability, such as epilepsy, neuropathic pain, and tinnitus are linked to genetic or experience-dependent reductions in KCNQ2/3 channel activity (*Biervert et al., 1998*; *Dedek et al., 2001*; *Kullmann, 2002*; *Wuttke et al., 2007*; *Li et al., 2013*). Pharmacological activation of KCNQ channels by retigabine, an FDA approved anti-epileptic drug that activates KCNQ2-5, or by SF0034, a potent KCNQ2/3 activator, prevent seizures, neuropathic pain and the development of tinnitus (*Gunthorpe et al., 2012*; *Li et al., 2013*; *Kalappa et al., 2015*). Because in vitro studies have shown that KCNQ2/3 channel activity is plastic and can be enhanced in response to increased neuronal activity (*Wu et al., 2008*; *Brown and Randall, 2009*; *Misonou, 2010*), it is possible that endogenous, non-pharmacologically driven recovery in KCNQ2/3 channel activity may provide resilience to KCNQ2/3-related pathology. However, such KCNQ2/3-mediated resilience mechanisms have not been observed in hyperexcitability-related disorders. Here, we report that natural, non-pharmacologically driven recovery in KCNQ2/3

channel activity is linked with the resilience to noise-induced tinnitus. This novel finding suggests a temporal window during which endogenous, intrinsic mechanisms can restore KCNQ2/3 channel activity and lead to tinnitus resilience. Elucidation of these mechanisms will unmask previously unknown plasticity mechanisms of KCNQ2/3 channel activity, and lead to new targets for drug development towards enhancing resilience to noise-induced tinnitus in humans.

KCNQ2/3 channels mediate the native neuronal M-type current (*Wang et al., 1998*), a slowly activating voltage-gated potassium current blocked by muscarinic acetylcholine (M) receptors (*Brown and Adams, 1980*). KCNQ2/3 currents are strongly inhibited not only by activation of M receptors, but also by activation of other G protein-coupled receptors that reduce membrane phosphatidylinositol-(4,5)-bisphosphate (PIP2) levels, which is the major determinant in KCNQ channel gating (*Marrion, 1997*; *Zhang et al., 2003*). Our previous studies have revealed the important role of cholinergic activity and M receptors, namely M1 and M3, in fusiform cell synaptic plasticity (*Zhao and Tzounopoulos, 2011*). Importantly, because noise exposure increases basal cholinergic activity in the DCN (*Jin et al., 2006*; *Kaltenbach and Zhang, 2007*; *Manzoor et al., 2013*), we propose that noise-induced increases and decreases (or recovery to baseline levels) in basal (tonic) M receptor signaling may underlie the bidirectional plasticity of KCNQ2/3 channel activity.

## Plasticity of HCN channels in brain disorders

HCN channels are non-inactivating cation channels that open at hyperpolarized voltages. Because of their voltage-dependent properties and their reversal potential at $-30$ mV, HCN channels not only depolarize the RMP, but also reduce membrane resistance and therefore stabilize the membrane voltage by opposing its alterations in response to synaptic inputs (*Pape, 1996*; *Robinson and Siegelbaum, 2003*; *Biel et al., 2009*). Therefore, changes in HCN channel activity affect intrinsic and synaptic excitability in opposing directions. This differential influence of HCN channels on synaptic and intrinsic neuronal excitability has implicated HCN plasticity in the vulnerability and resilience to several neurological disorders characterized by abnormal synaptic or intrinsic excitability, such as epilepsy, neuropathic pain, depression, and Parkinson's disease (*Chaplan et al., 2003*; *Biel et al., 2009*; *Chan et al., 2011*; *Emery et al., 2011*; *Friedman et al., 2014*). Although future studies are needed to test the potential causal relationship between reduction of HCN channel activity in fusiform cell and the resilience to tinnitus, our results show that reduction of HCN channel activity is associated with resilience to tinnitus and with normal levels of fusiform cell spontaneous firing activity (*Figures 4 and 5*). We suggest that reduced HCN channel activity in non-tinnitus mice prevents fusiform cell hyperactivity and contributes to tinnitus resilience by hyperpolarizing the RMP of fusiform cell (*Figures 4 and 5*). Moreover, decreased HCN channel activity increased the steady-state $R_{in}$ of fusiform cells (*Figures 4C and 5D*), which is expected to enhance responsiveness of fusiform cells to synaptic inputs. Therefore, reduction of HCN channel activity in fusiform cells may contribute to enhanced evoked activity and potentially to noise-induced hyperacusis—the perception of moderate-level sounds as intolerably loud.

Previous studies have shown that large increases in postsynaptic calcium lead to enhancement in HCN current amplitude via NMDA receptor (NMDAR) activation, CaMKII activation and increased HCN channel expression (*Fan et al., 2005*), while smaller increases in calcium through L-type calcium channels and/or activation of mGluRs and PKC activation lead to activity-dependent decreases in HCN expression (*Brager and Johnston, 2007*; *Chan et al., 2011*). Pharmacological enhancement of KCNQ channels with retigabine reduces spontaneous firing rate in fusiform cells (*Figure 7—figure supplement 1C,D*) and promotes a decrease in HCN current amplitude (*Figure 6D*). Therefore, we propose that decreases in spontaneous firing that may be caused by the enhancement in KCNQ2/3 channel activity lead to changes in intracellular calcium, which, in turn, may trigger a homeostatic mechanism that decreases HCN currents in an effort to normalize spontaneous spike rates. Immunohistochemical studies have shown that the HCN2 subunit is expressed in fusiform cells that lack HCN1 subunit expression (*Koch et al., 2004*), suggesting that HCN2 isoforms may mediate the noise-induced plasticity in fusiform cells. Therefore, we propose that manipulations that reduce HCN2 channel activity may serve as potential therapeutic path for preventing the development of tinnitus.

## Contribution of other conductances to vulnerability and resilience to tinnitus

Reduction of KCNQ2/3 currents is associated with increased spontaneous firing rate and tinnitus behavior 7 days after noise exposure (Li et al., 2013). However, at 4 days after noise exposure, fusiform cells show reduced KCNQ2/3 currents (Figure 3B) but without hyperactivity (Figure 8B). This lack of association between reduced KCNQ2/3 channel activity and neuronal hyperactivity can be explained by the hyperpolarized RMP of fusiform cells 4 days after noise exposure (Figure 8C). Because various levels of inwardly rectifying potassium channels ($K_{ir}$) set the diverse RMP of fusiform cells (Leao et al., 2012), we suggest that noise-induced increases in $K_{ir}$s may mediate this hyperpolarization of RMP and hence resilience to tinnitus. We conclude that reduction in KCNQ2/3 channel activity is a robust mechanism for triggering tinnitus, but its pathogenic effect is gated by the RMP of fusiform cells. Therefore, besides targeting KCNQ2/3 channel activators for preventing tinnitus, manipulations that hyperpolarize the fusiform cell RMP may also provide additional therapeutic approaches.

### Homeostatic plasticity and tinnitus

Fusiform cells from control, 4 days noise-exposed and 7 days non-tinnitus mice show similar spontaneous firing rates but with different combinations of ionic conductances (Figures 3, 5 and 8 and (Li et al., 2013)). These findings complement and extend previous experimental and theoretical work showing that similar neuronal output can result from multiple combinations of intrinsic and synaptic properties (Edelman and Gally, 2001; Prinz et al., 2004; Marder and Goaillard, 2006; Goaillard and Dufour, 2014; Ratte et al., 2014).

In accordance with this view, activity-dependent changes in conductances that affect neuronal excitability frequently trigger homeostatic, compensatory changes in different conductances, which result in constant neuronal output (Desai et al., 1999; LeMasson et al., 1993; O'Leary et al., 2010; O'Leary et al., 2013; O'Leary et al., 2014). Our results are consistent with such homeostatic mechanisms and highlight that recovery of KCNQ channel activity is associated with a reduction in HCN channel activity and the maintenance of normal spontaneous firing rates in non-tinnitus mice (Figures 3, 5 and (Li et al., 2013). Because homeostatic and coordinated regulation of potassium and HCN currents occurs in different species and neuronal circuits, such as dopaminergic neurons in rat susbstantia nigra pars compact, neurons of the lobster stomatogastric ganglion and octopus and MSO auditory brainstem neurons (MacLean et al., 2003; Oertel et al., 2008; Khurana et al., 2011; Amendola et al., 2012), we propose that coordinated plasticity of potassium and HCN channels may represent a general biophysical strategy for achieving neuronal homeostasis.

The fact that multiple molecular pathologies underlie hyperexcitability-related disorders, such as neuropathic pain and epilepsy, has led to the suggestion that drugs that simultaneously target more than one type of ion channels could treat these disorders more effectively (Goaillard and Dufour, 2014; Klassen et al., 2011; Ratte et al., 2014). Similarly, because plasticity of multiple conductances is involved in tinnitus, we propose that a combination of drugs that enhance KCNQ2/3 and reduce HCN channel activity represents a potent therapeutic approach that will enhance resilience and reduce vulnerability to tinnitus. Moreover, simultaneous pharmacological enhancement of KCNQ2/3 and reduction of HCN channel activity is expected to act synergistically in stabilizing spontaneous firing rates in fusiform cells. This synergistic effect may in turn reduce the required concentration for each individual drug to exert its effect and therefore may lead to increased potency and reduced toxicity.

## Materials and methods

### Mouse model of tinnitus

Animals were handled and sacrificed according to methods approved by the Institutional Animal Care and Use Committee of the University of Pittsburgh. ICR (CD-1) male and female mice were noise exposed at postnatal day P17–P20. After anesthetizing the mouse with 1–1.5% isoflurane, a pipette tip that was connected with the speaker was inserted into left ear canal of the mouse (unilateral noise exposure). Narrow bandpass noise with a 1 kHz bandwidth centered at 16 kHz was presented at 116 dB SPL (dB) for 45 min. For sham-exposed (control) mice, the procedures were identical with the noise-exposed mice but without noise presentation. For 4 days sham- and noise-exposure experiments, P20—P23 (instead of P17—P20) mice were used for sham or noise exposure, so that electrophysiological

recordings and behavioral assessments 4 days after noise exposure were performed on mice that had similar age with the mice that were used for experiments 7 days after sham or noise exposure.

## Gap detection

The gap detection paradigm (*Turner et al., 2006*; *Li et al., 2013*) was used for assessing behavioral evidence of tinnitus. Gap detection of sham- or noise-exposed mice was assessed before exposure, 4 days or 1 week (6–7 days) after sham or noise exposure. Detailed testing has been described in *Li et al. (2013)*. Briefly, the gap detection testing consists of two types of trials, gap trials and no-gap trials (*Figure 1A* top left). In gap trials, a sound gap was embedded in a narrow bandpass background sound (1 kHz bandwidth centered at 10, 12, 16, 20, 24, and 32 kHz presented at 70 dB), which was followed by the startle stimulus (20 ms white noise burst at 104 dB); a 50-ms sound gap was introduced 130 ms before the startle stimulus. No-gap trials were the same as gap trials, but that no gap was introduced in the background sound. Gap trials and no-gap trials were presented as paired stimuli for the same background sound frequency, and were delivered in an alternating manner. The startle response represents the waveform of the downward pressing force that the mouse applies onto the platform in response to the startle stimulus. The ability of a mouse to detect sound gap was quantified by the gap startle ratio, which is the ratio of the peak-to-peak value of the startle waveform (amplitude in Arbitrary Unit, AU) in gap trials over the peak-to-peak value of the startle waveform of the paired no-gap trials.

## Prepulse inhibition

Prepulse inhibition (PPI) is the inverse of gap detection and was tested together with gap detection before, 4 days, or 1 week after sham or noise exposure. PPI testing consists of prepulse trials and startle-only trials, which were delivered in an alternating manner (*Figure 1A*, bottom left). In prepulse trials, a brief non-startling sound (prepulse) of similar intensity as the background sound used in the gap detection test (50 ms, 70 dB bandpass sound with 1 kHz bandwidth centered at 10, 12, 16, 20, 24, and 32 kHz). The prepulse was presented 130 ms before the startle stimulus. Startle-only trials were similar to the prepulse trials, but no prepulse was delivered. PPI was quantified by the PPI ratio, which is the ratio of the peak-to-peak value of the startle waveform in prepulse trials over the peak-to-peak value of the startle waveform in startle-only trials.

## Gap detection and PPI analysis

For each sham- or noise-exposed mouse, gap detection and PPI ratios were averaged from three rounds of testing before and after sham or noise exposure. Each round of testing included 72 pairs of gap and no-gap trials for gap detection (6 background sound frequencies, 12 pairs for each frequency) and 30 pairs of prepulse and startle-only trials for PPI (6 prepulse sound frequencies, 5 pairs for each frequency). Gap startle and PPI ratios were analyzed separately. Maximum absolute amplitude of the startle response and root mean square of baseline movement preceding the startle response (RMS baseline) were measured with a Labview-based recording system. Mean and standard deviation of RMS baseline of gap, no-gap, prepulse, and startle-only trials were measured to assess the variability of baseline movement. Trials with RMS baseline amplitude above or below the mean $\pm$ 2.5 standard deviations were eliminated. When a trial was eliminated, its paired trial was also eliminated. For gap detection trials of the same background frequency, gap startle ratios were sorted in an ascending manner. To control for variability of gap startle ratios tested with the same frequency, ratios that showed an increase of more than 0.5 from the preceding value were excluded; the following values were also excluded. If more than 5 ratios were excluded within a frequency, the gap startle ratio for this frequency was not used. For background and prepulse sound of the same testing frequency, individual gap startle and PPI ratios were averaged and generated the average ratio for each round of testing. Average gap startle and PPI ratios of each frequency were then averaged across the three rounds of pre-exposure and post-exposure testing to generate the final ratio value. According to previous established criteria, gap startle ratios that were bigger than 0.9 before sham or noise exposure, or bigger than 1.1 after exposure were excluded (*Li et al., 2013*). Similarly, PPI ratios that were bigger than 1 before or after exposure were excluded (*Li et al., 2013*). Changes in gap startle ratio before and after exposure ($\Delta$ gap startle ratio) were calculated by subtracting the post-exposure ratio from the pre-exposure ratio for each testing frequency. The probability distribution of $\Delta$ gap startle ratios from all testing frequencies of sham-exposed mice was fitted with a Gaussian distribution, which permitted

the calculation of the mean (μ) and the standard deviation (δ) of the probability distribution (*Figure 1—figure supplement 1A*, *Figure 8—figure supplement 1A,B*), as described previously (*Li et al., 2013*). For evaluating the behavioral evidence of tinnitus, we calculated the point that is 2 standard deviations above the mean and used this value as the threshold (*Li et al., 2013*). Mice that presented Δ gap startle ratio higher than threshold value in at least one tested frequency were considered tinnitus mice (*Li et al., 2013*). To determine whether Δ gap startle ratios from 4 days noise-exposed mice were different from Δ gap startle ratios from 4 days sham-exposed mice, we calculated the fraction of Δ gap startle ratios that are above thresholds in 4 days sham- and noise-exposed mice, with thresholds ranging from 1 to −1 with 0.02 increments (*Figure 8—figure supplement 1C*).

## Auditory brainstem responses

Auditory brainstem response (ABR) thresholds were measured before, 4 days and 7 days after noise exposure following gap detection and PPI test. Measurements were conducted in a sound-attenuating chamber (ENV-022SD; Med Associates, St. Albans, VT, United States). Mice were anesthetized initially with 3% isoflurane in oxygen, and then maintained with 1–1.5%. To present the sound stimuli, a pipette tip was fixed to the end of a plastic tube (2.5 cm in length), which was attached to the speaker (CF-1; Tucker Davis Technologies, Alachua, FL, United States) and was inserted into the left ear canal. ABR thresholds were obtained for 1-ms clicks and 3-ms tone bursts of 10, 12, 16, 20, 24, and 32 kHz presented at a rate of 18.56/s in response to stimuli with different intensities (80 dB–15 dB, −5 dB step). Stimuli were produced using the System 3 software package from Tucker Davis Technologies. Evoked potentials were averaged 1024 times and filtered using a 300- to 3000-Hz bandpass filter. Wave I amplitude of each ABR response was measured as the peak-to-peak amplitude of the first wave that could be identified by eye from the baseline variation. Wave I amplitude for different stimulation intensities at 20 kHz, 24 kHz, and 32 kHz were averaged as response to high-frequency sounds (*Figure 2C,D*; *Figure 7C,D*).

## In vivo administration of KCNQ channel activators

For these experiments, ICR (CD-1) mice (P17—P20), both male and female mice were used. For experiments where administration of vehicle and KCNQ activators start 30 min after noise exposure, 20 noise-exposed mice were injected with KCNQ channel activators: 10 mice with retigabine as its dihydrochloride salt and 10 mice with flupirtine, another KCNQ channel activator (*Mackie and Byron, 2008*). 18 noise-exposed mice were treated with vehicle: 11 mice with 0.9% saline paired with the retigabine group and 7 mice with 30% propylene glycol, 5% Tween 80, 65% D5W paired with the flupirtine group. Prior to noise exposure, all mice were assessed for gap detection and PPI. For the KCNQ channel activator group, retigabine or flupirtine were administered 30 min after noise exposure via IP injection at a dose of 10 mg/kg. In the vehicle group, the same volume of vehicle solution was administered 30 min after noise exposure via IP injection. All mice were further administered with KCNQ channel activator or vehicle twice a day every 12 hr for 5 days. Gap detection and PPI were retested 24 hr after the final injection. Only mice from the retigabine and saline vehicle group were used for electrophysiological recordings (*Figure 6C,D* and *Figure 7—figure supplement 1C,D*). For experiments where injection of retigabine or saline started at the end of day 3 after noise exposure, injections continued 3 times a day every 8 hr for 3 days at a dose of 10 mg/kg. Gap detection and PPI were tested before noise exposure and 24 hr after the final injection. Retigabine dihydrochloride was obtained from Santa Cruz Biotechnology (LKT Laboratories, St. Paul, MN, United States) and Alomone (Jerusalem, Israel). Flupirtine maleate was obtained from Selleck Chemicals (Houston, TX, United States).

## Electrophysiological recordings

Coronal slices of the left DCN (210 μm) were prepared from control and noise-exposed mice (P24—P27). Immediately, after brain slices were prepared, they were incubated in normal artificial cerebral spinal fluid (ACSF) at 36°C for 1 hr and then at room temperature. Fusiform cells were visualized using an Olympus upright microscope under oblique illumination condenser equipped with a XC-ST30 CCD camera and analog monitor. Cells were identified based on their morphological and electrophysiological characteristics. The preparation of slices and the identification for fusiform cells have been described in detail previously (*Tzounopoulos et al., 2004*). The incubation as well as external

recording solution contained (in mM): 130 NaCl, 3 KCl, 1.2 $KH_2PO_4$, 2.4 $CaCl_2.2H_2O$, 1.3 $MgSO_4$, 20 $NaHCO_3$, 3 NaHEPES, and 10 D-glucose, saturated with 95% $O_2$/5% $CO_2$. DNQX (20 μM, AMPA and Kainate receptor antagonist, Abcam, Cambridge, MA, United States), strychnine (0.5 μM, glycine receptor antagonist, Sigma–Aldrich), SR95531 (20 μM, GABAa receptor antagonist, Abcam) were used to block glutamatergic, glycinergic as well as GABAergic synaptic transmission, respectively. XE991 (10 μM, KCNQ channel blocker, Abcam) was applied for blocking KCNQ currents. Tetrodotoxin (TTX, selective inhibitor for sodium channel, 0.5 μM, Abcam) was used to block spiking activity. ZD7288 (10 μM, blocker for HCN channel, Abcam) was used to block HCN channel activity. Recordings were performed at 34–37°C using an inline heating system (Warner Instruments, Hamden, CT, United States) with perfusion speed maintained (4–6 ml min$^{-1}$).

For whole-cell voltage and current clamp experiments, pipettes (3–5 MΩ) were filled with a $K^+$-based internal solution containing (in mM): 113 K-gluconate, 4.5 $MgCl_2$, 2.6 $H_2O$, 14 Tris-phosphocreatine, 9 HEPES, 0.1 EGTA, 4 $Na_2ATP$, 0.3 Tris-GTP, 10 Sucrose, pH 7.3, and 300 mOsmol. Liquid junction potential of −11 mV was corrected. Access resistance was monitored throughout the experiment from the size and shape of the capacitive transient in response to a 5-mV depolarization step. Recordings with access resistance larger than 15 MΩ were eliminated. Recordings were performed with Clampex 10.2 and Multiclamp 700B amplifier interfaced with Digidata 1440A data acquisition system (Axon Instruments). For whole-cell voltage clamp experiments, fast, slow capacitive currents as well as series resistance (Rs) were compensated (70%, bandwidth 15 kHz). In voltage clamp ramp experiments, $KH_2PO_4$ was removed from ACSF. External $CsCl_2$ (1 mM) and $CdCl_2$ (200 μM) were used to block hyperpolarization-activated cyclic nucleotide-gated channels (HCN, Ih channel) and calcium channels, respectively. All recording protocols, except for the gap-free recording in current clamp, were applied below 0.1 Hz to eliminate potential short-term plasticity effects. Spike parameters were analyzed from spontaneous spikes in whole-cell, voltage-follower mode recordings (current clamp, at I = 0; synaptic transmission was pharmacologically blocked). To assess spike properties, 20 consecutive spontaneous spikes were aligned at the negative peak and averaged. Spike threshold is the membrane potential at which the depolarization slope exceeds 10 V/s. Spike amplitude is the voltage difference between the spike threshold and the peak voltage of the spike. Depolarization and hyperpolarization slope indicate the maximum positive slope during the depolarization and minimum negative slope during hyperpolarization phase of the spike. Half height width is the width of the spike when voltage equals to (spike threshold + half of spike amplitude). Fast afterhyperpolarization (fAHP) is the voltage difference between spike threshold and negative peak of the spike. Resting membrane potential (RMP) was measured with whole-cell, voltage-follower mode recordings (current clamp, at I = 0) 5 min after TTX (0.5 μM) application. ZD7288 was applied after TTX for evaluating its effect on RMP. Input resistance ($R_{in}$) was measured in current clamp mode through current injection (−60 pA–60 pA, 20 pA step size, 2 s). Onset $R_{in}$ is the slope of the current–voltage (I–V) relationship of the average membrane voltage response of the initial 100 ms starting from the peak of the voltage response. Steady-state $R_{in}$ was calculated similarly to the onset $R_{in}$, but the last 250 ms of the voltage response were used. To quantify KCNQ2/3 currents, we measured the XE991-sensitive tail current amplitude in response to a voltage step to −50 mV from a holding potential of −30 mV, described in detail previously (*Li et al., 2013*). To measure HCN channel activity, we injected bias current to maintain the membrane potential at −75 mV. A hyperpolarizing current step (−550 pA, 2 s) was then used to activate HCN channels. The biggest membrane potential change from −75 mV ($V_{peak}$), and the membrane potential change at the end of hyperpolarizing current ($V_{ss}$) was used to calculate the sag ratio: Sag ratio = ($V_{peak}$−$V_{ss}$)/$V_{peak}$ × 100% (*Figure 5A*). Sag ratio before and after ZD7288 (15 min application) was subtracted and generated the ZD7288-sensitive sag ratio. In voltage-clamp ramp experiments, XE991-sensitive KCNQ currents elicited by slow voltage ramp (10 mV/s) were converted to conductance (G, nS) according to Ohm's law: G = I/(V−$V_r$). I (pA) is the current amplitude at the membrane potential V (mV), and $V_r$ is the reversal potential of potassium [−85.5 mV; (*Leao et al., 2012*)]. Conductance–voltage curves were then fitted with Boltzmann function to describe the voltage dependence of KCNQ activation (*Li et al., 2013*) (*Figure 3C,D,E*).

## Statistics

For data that were normally distributed (based on Liliefors test), we conducted Student's t-test or One-Way Analysis of Variance (ANOVA). Post-hoc analysis for one-way ANOVA was performed with

Tukey's least significant test. For non-normally distributed data, we performed non-parametric Wilcoxon rank sum test or Kruskal–Wallis test. Binomial test was used for comparing percentages of tinnitus mice in response to different experimental manipulations.

## Acknowledgements

We thank Geoffrey Zettell and Inga Kristaponyte for helping with some experiments assessing the behavioral evidence of tinnitus. This work was supported by the Department of Defense Peer Reviewed Medical Research Program Grant PR093405 Joint Warfighter Medical Research Program Grant W81XWH-14-1-0117 (to TT) and R01-DC007905 (to TT).

## Additional information

### Funding

| Funder | Grant reference | Author |
|---|---|---|
| U.S. Department of Defense (DOD) | PR093405 | Thanos Tzounopoulos |
| U.S. Department of Defense (DOD) | W81XWH-14-1-0117 | Thanos Tzounopoulos |
| National Institute on Deafness and Other Communication Disorders (NIDCD) | DC–DC007905 | Thanos Tzounopoulos |

The funders had no role in study design, data collection and interpretation, or the decision to submit the work for publication.

### Author contributions

SL, TT, Conception and design, Acquisition of data, Analysis and interpretation of data, Drafting or revising the article; BIK, Acquisition of data, Analysis and interpretation of data, Drafting or revising the article

### Ethics

Animal experimentation: Animals were handled, anesthetized and sacrificed according to methods approved by the University of Pittsburgh Institutional Animal Care and Use Committee. The approved IACUC protocol numbers that were employed for this study were: #14125118 and #14094011.

## Additional files

### Supplementary file

• Supplementary file 1. Spike parameters of fusiform cells from control and non-tinnitus mice. Spike threshold: control, n = 8, non-tinnitus, n = 8, p = 0.16; spike amplitude: control, n = 8, non-tinnitus, n = 8, p = 0.98; depolarization slope: control, n = 8, tinnitus, n = 8, p = 0.97; hyperpolarization slope: control, n = 8, tinnitus, n = 8, p = 0.07; half height width: control, n = 8, tinnitus, n = 8, p = 0.09; fast afterhyperpolarization (fAHP): control, n = 8, tinnitus, n = 8, p = 0.68).

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

## Appendix 1

# Detailed values for main Figures

*Figure 1B*. Control, high frequency: before exposure, 0.70 ± 0.01, after exposure: 0.68 ± 0.02, n = 21, p = 0.6; low frequency, before exposure, 0.67 ± 0.02, after exposure, 0.71 ± 0.02, n = 21, p = 0.2; tinnitus, high frequency: before exposure, 0.64 ± 0.02, after exposure: 0.82 ± 0.03, n = 11, p < 0.001; low frequency, before exposure, 0.66 ± 0.03, after exposure, 0.74 ± 0.03, p = 0.06, n = 11; non-tinnitus, high frequency: before exposure, 0.69 ± 0.03, after exposure: 0.71 ± 0.04, n = 10, p =0.6; low frequency, before exposure, 0.72 ± 0.02, after exposure, 0.64 ± 0.03, p = 0.07, n = 10.

*Figure 1C*. Control, high frequency, 0.56 ± 0.03, after exposure, 0.60 ± 0.02, n = 22, p = 0.25; low frequency, before exposure: 0.59 ± 0.02; after exposure: 0.60 ± 0.01, n = 22, p = 0.56; tinnitus, high frequency, before exposure, 0.52 ± 0.03, after exposure, 0.57 ± 0.02, n = 11, p = 0.18; low frequency, before exposure, 0.52 ± 0.04, after exposure, 0.59 ± 0.03, n = 11, p = 0.17; non-tinnitus, high frequency, before exposure, 0.54 ± 0.03, after exposure, 0.60 ± 0.03, n = 10, p = 0.36; low frequency, before exposure, 0.56 ± 0.04, after exposure, 0.54 ± 0.02, n = 10, p = 0.56.

*Figure 2B*. Before noise exposure: click, tinnitus, 38.5 ± 4.5 dB, n = 10, non-tinnitus, 39.5 ± 2.8 dB, n = 11, p = 0.84; 10 kHz, tinnitus, 38.2 ± 1.7 dB, n = 11, non-tinnitus, 36.8 ± 1.8 dB, n = 11, p = 0.59; 12 kHz, tinnitus, 34.0 ± 1.5 dB, n = 10, non-tinnitus, 33.2 ± 1.9 dB, n = 11, p = 0.74; 16 kHz, tinnitus, 38.0 ± 2.3 dB, n = 10, non-tinnitus, 37.7 ± 2.8 dB, n = 11, p = 0.94; 20 kHz, tinnitus, 25.0 ± 2.5 dB, n = 11, non-tinnitus, 30.5 ± 1.9 dB, n = 11, p = 0.74; 24 kHz, tinnitus, 38.2 ± 1.7 dB, n = 11, non-tinnitus, 36.8 ± 1.8 dB, n = 11, p = 0.59; 32 kHz, tinnitus, 39.0 ± 1.9 dB, n = 5, non-tinnitus, 35.5 ± 2.8 dB, n = 10, p = 0.43; After noise exposure: click, tinnitus, 33.8 ± 2.3 dB, n = 8, non-tinnitus, 33.1 ± 1.3 dB, n = 8, p = 0.81; 10 kHz, tinnitus, 67.2 ± 4.0 dB, n = 9, non-tinnitus, 65.0 ± 6.1 dB, n = 8, p = 0.76; 12 kHz, tinnitus, 63.3 ± 4.6 dB, n = 9, non-tinnitus, 56.9 ± 5.7 dB, n = 8, p = 0.39; 16 kHz, tinnitus, 68.3 ± 4.3 dB, n = 10, non-tinnitus, 62.5 ± 6.3 dB, n = 8, p = 0.45; 20 kHz, tinnitus, 51.0 ± 2.4 dB, n = 5, non-tinnitus, 55.0 ± 1.3 dB, n = 8, p = 0.14; 24 kHz, tinnitus, 63.3 ± 1.7 dB, n = 9, non-tinnitus, 60.7 ± 1.3 dB, n = 7, p = 0.43; 32 kHz, tinnitus, 54.4 ± 2.6 dB, n = 8, non-tinnitus, 59.0 ± 2.4 dB, n = 5, p = 0.25).

*Figure 2C*. (35 dB: tinnitus, 759.0 ± 97.0 nV, n = 22, non-tinnitus, 661.7 ± 53.6 nV, n = 28, p = 0.36; 40 dB: tinnitus, 945.0 ± 109.1 nV, n = 24, non-tinnitus, 768.7 ± 52.8 nV, n = 29, p = 0.13; 45 dB: tinnitus, 1030.9 ± 112.3 nV, n = 24, non-tinnitus, 825.5 ± 51.1 nV, n = 28, p = 0.09; 50 dB: tinnitus, 1103.6 ± 112.4 nV, n = 22, non-tinnitus, 1057.8 ± 88.5 nV, n = 25, p = 0.75; 55 dB: tinnitus, 1166.2 ± 147.4 nV, n = 21, non-tinnitus, 1029.5 ± 92.2 nV, n = 21, p = 0.45; 60 dB: tinnitus, 1456.2 ± 257.8 nV, n = 14, non-tinnitus, 1157.1 ± 91.4 nV, n = 21, p = 0.22; 65 dB: tinnitus, 1529.7 ± 236.0 nV, n = 12, non-tinnitus, 1210.2 ± 125.7 nV, n = 17, p = 0.21).

*Figure 2D*. (55 dB: tinnitus, 456.7 ± 286.9 nV, n = 5, non-tinnitus, 330.1 ± 82.7 nV, n = 7, p = 0.64; 60 dB: tinnitus, 494.1 ± 200.1 nV, n = 5, non-tinnitus, 358.5 ± 94.5 nV, n = 8, p = 0.5; 65 dB: tinnitus, 402.0 ± 112.7 nV, n = 10, non-tinnitus, 270.0 ± 59.7 nV, n = 8, p = 0.35; 70 dB, tinnitus, 603.7 ± 310.0 nV, n = 5; non-tinnitus, 529.4 ± 114.3 nV, n = 4, p = 0.56).

*Figure 7B*. Before noise exposure: click, noise-exposed + saline, 33.8 ± 0.8 dB, n = 8, noise-exposed + retigabine, 30.0 ± 2.2 dB, n = 6, p = 0.10; 10 kHz, noise-exposed + saline, 38.8 ± 1.8 dB, n = 8, noise-exposed + retigabine, 34.3 ± 1.7 dB, n = 7, p = 0.10; 12 kHz, noise-exposed + saline, 30.7 ± 3.8 dB, n = 7, noise-exposed + retigabine, 29.3 ± 4.0 dB, n = 7, p = 0.80; 16 kHz, noise-exposed + saline, 30.6 ± 1.0 dB, n = 9, noise-exposed + retigabine, 25.7 ± 2.8 dB, n = 7, p = 0.16; 20 kHz, noise-exposed + saline, 26.1 ± 5.1 dB, n = 9, noise-exposed + retigabine, 20.0 ± 4.1 dB, n = 4, p = 0.48; 24 kHz, noise-exposed + saline, 26.9 ± 2.5 dB, n = 8, noise-exposed + retigabine, 21.0 ± 3.0 dB, n = 5, p = 0.16; 32 kHz, noise-exposed + saline, 25.0 ± 2.0 dB, n = 4, non-tinnitus, 23.8 ± 2.4 dB, n = 4, p = 0.70; After noise exposure: click, noise-exposed + saline, 31.0 ± 1.9 dB,

n = 5, noise-exposed + retigabine, 32.9 ± 2.6 dB, n = 7, p = 0.61; 10 kHz, noise-exposed + saline, 53.8 ± 7.2 dB, n = 4, noise-exposed + retigabine, 54.0 ± 4.6 dB, n = 10, p = 0.98; 12 kHz, noise-exposed + saline, 51.3 ± 5.5 dB, n = 4, noise-exposed + retigabine, 49.4 ± 5.5 dB, n = 8, p = 0.84; 16 kHz, noise-exposed + saline, 60.0 ± 5.7 dB, n = 5, noise-exposed + retigabine, 56.3 ± 4.9 dB, n = 8, p = 0.63; 20 kHz, noise-exposed + saline, 56.0 ± 1.9 dB, n = 5, noise-exposed + retigabine, 56.0 ± 2.1 dB, n = 10, p = 0.25; 24 kHz, noise-exposed + saline, 57.5 ± 1.4 dB, n = 4, noise-exposed + retigabine, 59.3 ± 2.0 dB, n = 7, p = 0.56; 32 kHz, noise-exposed + saline, 56.7 ± 1.1 dB, n = 6, noise-exposed + retigabine, 52.0 ± 2.0 dB, n = 5, p = 0.15.

*Figure 7C*. 35 dB, noise-exposed + saline, 815.2 ± 68.3 nV, n = 15, noise-exposed + retigabine, 826.1 ± 76.8 nV, n = 14, p = 0.92; 40 dB, noise-exposed mice + saline, 947.8 ± 81.0 nV, n = 16, noise-exposed + retigabine, 1054.5 ± 104.8 nV, n = 14, p = 0.42; 45 dB, noise-exposed + saline, 963. 2 ± 69.9 nV, n = 14, noise-exposed + retigabine, 1163.5 ± 129.1 nV, n = 13, p = 0.16; 50 dB, noise-exposed + saline, 1022.6 ± 64.3 nV, n = 16, noise-exposed + retigabine, 1207.6 ± 99.9 nV, n = 14, p = 0.12; 55 dB, noise-exposed + saline, 1247.1 ± 76.7 nV, n = 13, noise-exposed + retigabine, 1266.6 ± 134.5 nV, n = 12, p = 0.90; 60 dB, 1168.4 ± 74.8 nV, n = 9, noise-exposed + retigabine, 1392.2 ± 170.8 nV, n = 10, p = 0.26; 65 dB, noise-exposed + saline, 1097.1 ± 117.0 dB, n = 5, noise-exposed + retigabine, 1183.3 ± 156.0 nV, n = 5, p = 0.67).

*Figure 7D*. 55 dB, noise-exposed + saline, 240.0 ± 41.7 nV, n = 6, noise-exposed + retigabine, 320.0 ± 61.6 nV, n = 10, p = 0.41; 60 dB, noise-exposed mice + saline, 340.4 ± 32.7 nV, n = 16, noise-exposed + retigabine, 312.5 ± 49.9 nV, n = 20, p = 0.66; 65 dB, noise-exposed + saline, 471.9 ± 52.5 nV, n = 13, noise-exposed + retigabine, 372.7 ± 56.5 nV, n = 14, p = 0.21; 70 dB, noise-exposed + saline, 401.3 ± 78.9 nV, n = 8, noise-exposed + retigabine, 458.7 ± 56.7 nV, n = 12, p = 0.55; 75 dB, noise-exposed + saline, 548.6 ± 156.6 nV, n = 6, noise-exposed + retigabine, 590.4 ± 72.7 nV, n = 5, p = 0.83).

