## [Decision Letter]

[Editors’ note: this article was originally rejected after discussions between the reviewers, but the authors submitted for further consideration.]

Thank you for sending your work entitled “Noise-Induced Plasticity of Kv7.2/3 (KCNQ2/3) and HCN Channels Underlies Vulnerability and Resilience to Tinnitus” for consideration at *eLife*. Your article has been evaluated by Eve Marder (Senior Editor), Gary Westbrook (Reviewing Editor), and two reviewers. The Reviewing editor and the other reviewers discussed their comments before they reached a decision.

Although this study is generally well-executed and provides results relevant to a major public health issue, the reviewers felt that significant additional information is needed to support the main conclusions. Thus per *eLife* policy not to require substantial additional experiments, we cannot consider the manuscript further at this time.

This study addresses the question of why intense noise exposure leads to tinnitus in some mice but not others. The basis of this work is that all noise-exposed animals show shifts in hearing threshold, but only a proportion of them show increased spontaneous firing rates in fusiform cells in the dorsal cochlear nucleus (DCN). Previously, Li, Choi, and Tzounopoulos (PNAS, 2013) showed that KCNQ2/3 (M) currents are reduced in DCN fusiform cells of mice exhibiting signs of tinnitus seven days after intense noise exposure but not in noise-exposed asymptomatic mice. A clever behavioral task is used to separate whether mice can detect a silent gap in background noise preceding a startle stimulus. “Tinnitus” mice are defined as those that cannot detect a silent gap without apparent differences in noise-induced hearing loss as measured by a prepulse inhibition (PPI) protocol.

The authors report that KCNQ2/3 currents are reduced in the fusiform cells of all noise-exposed mice four days after noise exposure, even though statistically nearly half of these mice would not have gone on to develop tinnitus. This suggests that KCNQ2/3 currents return to pre-noise exposure levels in tinnitus-resistant mice between four and seven days after noise exposure, providing a possible molecular explanation for why some noise-exposed mice develop tinnitus while others do not. In addition, the authors also report that HCN currents and resting membrane potentials are reduced in tinnitus-resistant mice seven days after noise exposure, but not in tinnitus mice.

Major issues:

1) The extent of damage from the noise exposure is not clear in the study. It seems important to know if the differences between “tinnitus” and “non-tinnitus” mice results from differences in the extent of noise damage to hair cells, or differences in the response of the DCN to comparable damage, i.e. do responses to noise-exposure reflect different plastic responses of the DCN to the same injury, or differences in the extent of the upstream injury. For example, could the protective effect of KCNQ activation result from protection of the cochlea and the plasticity differences observed reflect a uniform response to different extents of damage? At present, this distinction seems to depend solely on the PPI measurement. The selectivity of deficits in Gap startle ratio vs. paired pulse inhibition are helpful and important, but the conclusions in the subsection “Mouse model of tinnitus allows for behavioral separation of noise-exposed mice with either vulnerability or resilience to tinnitus” seem too strong. The authors claim their results rule out impairments in temporal processing or ability to hear background noise. Selective deficits in temporal processing of e.g. offset responses could explain the results, and not relate to tinnitus. Caveats of the model should be made clear. One possible resolution would be to perform DPOAEs or ABRs in a group of animals in which they also assess “tinnitus” versus “non-tinnitus” DCN phenotype, following e.g.: Kujawa, S. G., & Liberman, M. C. (2009), Adding insult to injury: cochlear nerve degeneration after “temporary” noise-induced hearing loss, The Journal of Neuroscience, 29(45), 14077-14085.

2) What is the evidence that that a selective effect of noise exposure on Gap detection is a reliable indication that an animal has tinnitus? Because tinnitus is a percept, and its physiological manifestations have been elusive, it is not at all clear at the outset how one knows if an animal is experiencing tinnitus.

3) The sole metric used for assessing KCNQ2/3 currents is the amplitude of the tail currents resulting from a voltage step from -30 mV to -50 mV (e.g. Figure 2). A more informative measure would be to generate voltage activation curves as the authors did in their 2013 paper (PNAS Figure 3). This would provide more direct evidence about whether all noise-exposed mice exhibit the same mechanism for KCNQ2/3 current reduction (e.g. change in Vhalf versus Gmax) or if a subtle but meaningful dichotomy exists between tinnitus-resistant and susceptible mice at day 4.

4) Evidence that recovery of KCNQ2/3 currents 4 to 7 days after noise exposure provides resistance to the development of tinnitus is based on a correlation and does not directly demonstrate that KCNQ currents are important. To more directly assess whether an increase in KCNQ2/3 currents between days 4 and 7 drives resilience to tinnitus, the authors could test whether retigabine treatment initiated at day 4 instead of 30 minutes after noise exposure is sufficient to reduce the incidence of tinnitus at day 7.

5) Figure 5–figure supplement 1 suggests that KCNQ activator treatment reduces the percentage of mice that develop tinnitus. In Figure 5–figure supplement 1B, however, the evidence that treated mice do not exhibit an increased gap startle ratio rests on a p value of 0.06. This raises the serious concern that different results in a single mouse would change the interpretation of this entire experiment. How sensitive are the results in Figure 5–figure supplement 1A to the threshold used to assign mice to the tinnitus group?

6) The Discussion seems overly long and repetitive, but does not really explain the underlying signaling mechanisms involved in the “homeostatic” changes in subthreshold currents. Specifically, what predisposes a particular animal to have tinnitus or to be tinnitus-resistant. Given the information available about modulation of M current by PIP2, and PKC-CaM, at least these issues should be discussed in more detail.

[Editors’ note: what now follows is the decision letter after the authors submitted for further consideration.]

Thank you for resubmitting your work entitled “Noise-Induced Plasticity of KCNQ2/3 and HCN Channels Underlies Vulnerability and Resilience to Tinnitus” for further consideration at *eLife*. Your revised article has been favorably evaluated by Gary Westbrook (Reviewing Editor) and two reviewers. The manuscript has been improved but there are just a few remaining issues that need to be addressed before final acceptance, as outlined below:

The authors have performed additional experiments and analyses that, for the most part, adequately address the concerns raised in the previous reviews. The addition of new data, particularly the ABRs and the effectiveness of retigabine treatment when started 4 days after noise exposure, significantly strengthen support for the authors' main conclusions.

Given the broad readership of *eLife*, it would be helpful to include more discussion of the strengths and weaknesses of gap-detection as an assay of tinnitus. The authors state: “There are several review articles discussing advantages and disadvantages in using this behavioral paradigm, which are outside the scope of this study.” However, we think a few sentences in the Introduction and Discussion on the support for and arguments against the gap-detection model of tinnitus will be very helpful for the general reader. Without some discussion on this point I suspect a lot of non-auditory readers will be wondering “how can you know if a mouse is experiencing tinnitus?” Some discussion, even if it ostensibly points out weaknesses of the model, will reassure readers and put the paper in context.

---

## [Author Response]

[Editors’ note: the author responses to the first round of peer review follow.]

We have performed additional experiments and analysis; included 2 new figures (Figure 2 and Figure 7); added new panels on pre-existing figures (Figure 3 and Figure 7—figure supplement 1); increased the number of experiments on previous figures (Figure 5); and edited the text according to the editorial and reviewers’ comments.

*1) The extent of damage from the noise exposure is not clear in the study. It seems important to know if the differences between “tinnitus” and “non-tinnitus” mice results from differences in the extent of noise damage to hair cells, or differences in the response of the DCN to comparable damage, i.e. do responses to noise-exposure reflect different plastic responses of the DCN to the same injury, or differences in the extent of the upstream injury. For example, could the protective effect of KCNQ activation result from protection of the cochlea and the plasticity differences observed reflect a uniform response to different extents of damage? At present, this distinction seems to depend solely on the PPI measurement. The selectivity of deficits in Gap startle ratio vs. paired pulse inhibition are helpful and important, but the conclusions in the subsection “Mouse model of tinnitus allows for behavioral separation of noise-exposed mice with either vulnerability or resilience to tinnitus” seem too strong. The authors claim their results rule out impairments in temporal processing or ability to hear background noise. Selective deficits in temporal processing of e.g. offset responses could explain the results, and not relate to tinnitus. Caveats of the model should be made clear. One possible resolution would be to perform DPOAEs or ABRs in a group of animals in which they also assess “tinnitus” versus “non-tinnitus” DCN phenotype, following e.g.: Kujawa, S. G., & Liberman, M. C. (2009), Adding insult to injury: cochlear nerve degeneration after “temporary” noise-induced hearing loss, The Journal of Neuroscience, 29(45), 14077-14085*.

We agree that prepulse inhibition (PPI) measurements are not sufficient to address the issues raised by the reviewer regarding the potential role of differential injury between tinnitus and non-tinnitus mice upstream to the DCN. As suggested by the reviewers, to address these issues, we performed auditory brainstem response (ABR) measurements, analysis of hearing thresholds and analysis of wave I amplitude of the ABR. Please note that we had already performed these measurements in the same animals included in Figure 1, and therefore these results match the behavioral gap detection and PPI experiments included in Figure 1. Briefly, consistent with our PPI results, tinnitus and non -tinnitus mice displayed similar hearing thresholds before and after noise-exposure, as evidenced by similar ABR thresholds (Figure 2). Because similar ABR thresholds may not reveal differences in suprathreshold amplitudes of wave I, which would reflect differential degeneration of the auditory nerve ([38], J. of Neurosci.; [25], J. of Neurophys.), we compared the amplitude of the wave I in response to suprathreshold sounds between tinnitus and non-tinnitus mice before and after noise exposure. Whereas wave I amplitudes were reduced after noise-exposure for all noise-exposed mice, tinnitus and non-tinnitus mice showed no differences in wave I amplitude, suggesting an indistinguishable damage of afferent nerve terminals between these two groups (Figure 2). Taken together, our results suggest that neither differential noise- induced hearing threshold shifts nor differential auditory nerve damage can explain the behavioral differences between tinnitus and non-tinnitus mice. We included these data in the new Figure 2 and edited the text accordingly.

*2) What is the evidence that that a selective effect of noise exposure on Gap detection is a reliable indication that an animal has tinnitus? Because tinnitus is a percept, and its physiological manifestations have been elusive, it is not at all clear at the outset how one knows if an animal is experiencing tinnitus*.

We agree that it is not clear to know with certainty whether a mouse experiences tinnitus. Given the subjective nature of tinnitus perception, there is not any objective behavioral paradigm in either mice or humans that provides objective evidence of tinnitus. Despite these inherent limitations of studying tinnitus perception, the gap detection paradigm has been used by a large number of leading tinnitus neuroscientists including Drs. Caspary ([65], J. of Neurosci.), Kilgard ([21], Nature), Shore ([37], J. of Neurosci.), to mention a few. There are several review articles discussing advantages and disadvantages in using this behavioral paradigm, which are outside the scope of this study. Importantly, this paradigm has revealed extremely important findings for the neurobiology and treatment of tinnitus. In several cases, these findings have also led to ongoing clinical trials. For example the Kilgard study on the role of vagus nerve stimulation paired with tones for treating tinnitus was based on this paradigm ([21], Nature), and is currently under clinical trial. Our studies, also based on the gap detection paradigm ([42], PNAS; [32], Journal of Neuroscience), have revealed novel KCNQ2/3 channel activators for the treatment of tinnitus and epilepsy and are moving toward clinical trials. We therefore believe that the gap detection paradigm is a useful behavioral assay for elucidating tinnitus mechanisms.

*3) The sole metric used for assessing KCNQ2/3 currents is the amplitude of the tail currents resulting from a voltage step from -30 mV to -50 mV (e.g.*
Figure 2*). A more informative measure would be to generate voltage activation curves as the authors did in their 2013 paper (PNAS*
Figure 3*). This would provide more direct evidence about whether all noise-exposed mice exhibit the same mechanism for KCNQ2/3 current reduction (e.g. change in Vhalf versus Gmax) or if a subtle but meaningful dichotomy exists between tinnitus-resistant and susceptible mice at day 4*.

We agree and performed additional experiments to measure the voltage dependence of KCNQ currents 4 days after noise exposure. Briefly, a Boltzmann fit of the KCNQ2/3 conductance–voltage (G–V) function showed that the G_max_ of KCNQ2/3 currents was not different between sham- and 4 days noise-exposed mice (Figure 3), but the V_1/2_ was shifted to more depolarized potentials in the 4 days noise-exposed mice (Figure 3; Methods). These results suggest that the reduction of KCNQ2/3 currents in 4 days noise-exposed mice is due to a depolarizing shift in the V _1/2_ of KCNQ2/3 channels, which is mechanistically similar to the reduction of KCNQ2/3 currents in tinnitus mice when assessed 7 days after noise exposure ([42], PNAS) We included these data in the revised Figure 3 and edited the text accordingly.

*4) Evidence that recovery of KCNQ2/3 currents 4 to 7 days after noise exposure provides resistance to the development of tinnitus is based on a correlation and does not directly demonstrate that KCNQ currents are important. To more directly assess whether an increase in KCNQ2/3 currents between days 4 and 7 drives resilience to tinnitus, the authors could test whether retigabine treatment initiated at day 4 instead of 30 minutes after noise exposure is sufficient to reduce the incidence of tinnitus at day 7*.

We agree and performed additional behavioral experiments to address this issue. As suggested by the reviewer, we tested whether retigabine application initiated at day 4, instead of 30 min after noise exposure, is sufficient to reduce the incidence of tinnitus at day 7. Application of retigabine at day 4 reduced significantly the percentage of mice that developed tinnitus (Figure 3; Methods), suggesting that the recovery of KCNQ2/3 currents between day 4 and 7 is crucial for tinnitus resilience. We included these data in the revised Figure 3 and edited the text accordingly.

*5) Figure 5–figure supplement 1 suggests that KCNQ activator treatment reduces the percentage of mice that develop tinnitus. In Figure 5–figure supplement 1B, however, the evidence that treated mice do not exhibit an increased gap startle ratio rests on a p value of 0.06. This raises the serious concern that different results in a single mouse would change the interpretation of this entire experiment. How sensitive are the results in Figure 5–figure supplement 1A to the threshold used to assign mice to the tinnitus group*?

We are confident that KCNQ activator treatment reduces the percentage of mice that develop tinnitus, which is shown in Figure 5–figure supplement 1A. Moreover, these results reproduce our previous results published in two different manuscripts ([42], PNAS; [32], Journal of Neuroscience). The threshold used to assign mice to tinnitus group affects the outcome of the experiment, but it is established for every different behavioral experiment by using objective quantitative criteria. Particularly, changes in gap startle ratio before and after exposure (Δ gap startle ratio) were calculated by subtracting the post-exposure ratio from the pre-exposure ratio for each testing frequency. The probability distribution of Δ gap startle ratios from all testing frequencies of sham-exposed mice was fitted with a Gaussian distribution, which permitted the calculation of the mean (µ) and the standard deviation (δ) of the probability distribution (Figure 1—figure supplement 1, Figure 8—figure supplement 1), as described previously ([42], PNAS). For evaluating the behavioral evidence of tinnitus, we calculated the point that is 2 standard deviations above the mean and used this value as the threshold (Li eat al., 2013). Mice that presented Δ gap startle ratio higher than threshold value in at least one tested frequency were considered tinnitus mice (Li eat al., 2013, PNAS).

*6) The Discussion seems overly long and repetitive, but does not really explain the underlying signaling mechanisms involved in the “homeostatic” changes in subthreshold currents. Specifically, what predisposes a particular animal to have tinnitus or to be tinnitus-resistant. Given the information available about modulation of M current by PIP2, and PKC-CaM, at least these issues should be discussed in more detail*.

According to the reviewers’ comments the Discussion has been shortened by removing repetitions and one section, entitled DCN spontaneous firing rates and tinnitus induction, which is now included in the Introduction. Moreover, based on the reviewers’ comment, we now discuss the possibility that noise-induced increases and decreases (or recovery to baseline levels) in basal muscarinic acetylcholine (M) receptor signaling and PIP2 levels may underlie the bidirectional plasticity of KCNQ2/3 channel activity in response to noise exposure. This is included in the last paragraph of the Discussion section entitled “Plasticity of KCNQ2/3 channels in hyperexcitability-related brain disorders”.

[Editors’ note: the author responses to the re-review follow.]

*Given the broad readership of* eLife*, it would be helpful to include more discussion of the strengths and weaknesses of gap-detection as an assay of tinnitus. The authors state: “There are several review articles discussing advantages and disadvantages in using this behavioral paradigm, which are outside the scope of this study.” However, we think a few sentences in the Introduction and Discussion on the support for and arguments against the gap-detection model of tinnitus will be very helpful for the general reader. Without some discussion on this point I suspect a lot of non-auditory readers will be wondering “how can you know if a mouse is experiencing tinnitus?” Some discussion, even if it ostensibly points out weaknesses of the model, will reassure readers and put the paper in context*.

We discussed the strengths and the weaknesses of the tinnitus animal model. We added a new section in the Discussion entitled “Gap Detection in Tinnitus Animal Models and in Humans with Tinnitus”. This additional section is the first section of the Discussion.